# Data augmentation using Variational Autoencoders for improvement of respiratory disease classification

**Jane Saldanha**[1], **Shaunak Chakraborty**[2], **Shruti Patil**[1]*, **Ketan Kotecha**[1], **Satish Kumar**[1], **Anand Nayyar**[3]

**1** Symbiosis Centre for Applied Artificial Intelligence, Symbiosis Institute of Technology, Symbiosis International (Deemed University), Pune, Maharashtra, India, **2** Dept. of Computer Science, Symbiosis Institute of Technology, Symbiosis International (Deemed University), Pune, Maharashtra, India, **3** Graduate School (Computer Science), Faculty of Information Technology, Duy Tan University, Da Nang, Vietnam

* shruti.patil@sitpune.edu.in

**Data Availability Statement:** Data can be accessed through the following link: https://bhichallenge.med.auth.gr/ICBHI_2017_Challenge.

## Abstract

Computerized auscultation of lung sounds is gaining importance today with the availability of lung sounds and its potential in overcoming the limitations of traditional diagnosis methods for respiratory diseases. The publicly available ICBHI respiratory sounds database is severely imbalanced, making it difficult for a deep learning model to generalize and provide reliable results. This work aims to synthesize respiratory sounds of various categories using variants of Variational Autoencoders like Multilayer Perceptron VAE (MLP-VAE), Convolutional VAE (CVAE) Conditional VAE and compare the influence of augmenting the imbalanced dataset on the performance of various lung sound classification models. We evaluated the quality of the synthetic respiratory sounds' quality using metrics such as Fréchet Audio Distance (FAD), Cross-Correlation and Mel Cepstral Distortion. Our results showed that MLP-VAE achieved an average FAD of 12.42 over all classes, whereas Convolutional VAE and Conditional CVAE achieved an average FAD of 11.58 and 11.64 for all classes, respectively. A significant improvement in the classification performance metrics was observed upon augmenting the imbalanced dataset for certain minority classes and marginal improvement for the other classes. Hence, our work shows that deep learning-based lung sound classification models are not only a promising solution over traditional methods but can also achieve a significant performance boost upon augmenting an imbalanced training set.

## 1. Introduction

The inception of deep learning has paved the way for many breakthroughs in science, medicine, and engineering. Deep learning is rapidly developing in the field of acoustics, providing many compelling solutions to problems like Automatic Speech Recognition [1], sound synthesis [2], acoustic scene classification [3], generative music [4], acoustic event detection [5], and many more. The increase in access to data generated by health care and the development of deep learning techniques has proved to be prosperous. Deep Learning can be used to unravel

**Funding:** Symbiosis International (Deemed University) has provided the financial support for the manuscript APC and infrastructure support for implementing the proposed work. The funder had played no role in study design, data collection and analysis, decision to publish, or preparation of the manuscript.

**Competing interests:** The authors have declared that no competing interests exist.

the clinically pertinent information from the dataset provided by hospitals, which can be then used for decision making, treatment, and prevention of health conditions.

For efficacious training of neural networks, sufficient data is required. In the case of a low data scheme, the parameters are underdetermined, and the networks generalize poorly. Data Augmentation techniques enhance the generalizability of the deep learning model by using the original data efficiently. Standard data augmentation techniques like random translations, flips, and rotations, and the addition of Gaussian noise generates authentic but very little data.

Variational Autoencoders have been used to generate synthetic samples and improve the performance of the classifiers [6, 7]. The creation of synthetic data for minority classes can be useful for many reasons. The first reason for creating synthetic samples is to avoid the usage of original samples for privacy reasons. It is highly possible that working directly on the database containing the original samples can be misused or breached. The second reason is oversampling of the minority classes. In real-life scenarios, there is always a possibility of certain classes being underrepresented.

Respiratory diseases are a pathological condition that affects the tissues and organs which consequently makes gas exchange difficult. Some of the factors contributing to respiratory diseases are air pollution, tobacco smoke, occupational chemicals, dust, etc [8]. Respiratory diseases are one of the leading causes of death and disability in the world. Improving respiratory health not only entails reducing tobacco consumption or controlling the unhealthy air at the workplace but also involves strengthening the health care system and detection of respiratory disease via breathing sounds is a compelling addition.

## 2. Literature survey

### 2.1 Literature review on automated respiratory sounds auscultation algorithms

The development of automated respiratory sound auscultation algorithms requires the procurement of respiratory sounds either through various devices or publicly accessible databases, annotating these audios, pre-processing, followed by feature extraction and classification of these audios.

The acquisition of respiratory sounds from patients at clinics is achieved using various equipment like microphones or stethoscopes. The development of mobile technology has made it possible to acquire clinical samples and diagnoses. [9] proposed a modified stethoscope to record lung sounds. Further, the introduction of mHealth applications has made it possible to monitor vital organs, track treatment progress and send patient records to hospitals. Few readily available respiratory sounds databases include the Multimedia Database MARS (Marburg Respiratory Sounds), RALE Repository (Respiratory Acoustic Laboratory Environment), ICBHI (International Conference on Biomedical and Health Informatics) 2017 Scientific Challenge Dataset, etc. The Marburg Respiratory Sounds Database has over 5000 audio recordings from patients with COPD, bronchial Asthma, lung fibrosis and Pneumonia, along with clinical parameters such as lung function, X-ray findings and laboratory results. The RALE Repository provides a collection of sounds recorded by a stethoscope from the chest of patients. It was created for medical students to gain experience in lung sounds auscultation. The ICBHI Respiratory Sounds Dataset contains 5.5 hours of respiratory sounds obtained from 128 patients belonging to 8 different classes (1 healthy + 7 diseases).

The development of computerized lung sound auscultation algorithms requires annotated audio data. These annotations are valuable in pre-processing steps such as the segmentation of lung sounds and for the development of supervised learning algorithms. Respiratory sound annotation software makes it easier for medical experts to annotate lung sounds by providing

an intuitive interface for quickly identifying respiratory cycles and valuable sounds in the waveform such as wheezes and crackles. The development of Graphical User Interface (GUI) based applications such as LungSounds@UA has made it easier for users to interact with multi-media databases that store respiratory sounds and associated clinical parameters.

Pre-processing the acquired audio data is an essential step before feeding the data to a classification model. This is because the pre-processing techniques applied to the data affect the training efficiency and performance of a machine/deep learning model. Neural networks can represent any function. However, it may not effectively learn any input unless the correct pre-processing techniques have been used. Some of the pre-processing methods used for lung sounds include denoising, segmentation, normalization, resampling, spectrogram conversion, padding, trimming, etc. Lung sounds acquired from various devices are often corrupted by different noise sources such as heartbeats, speech, electrical interference, etc. [10] discusses denoising filters such as Butterworth Bandpass filters, wavelet transform based filters, FIR filters, etc. and compares their efficacies in eliminating certain types of noise from lung sounds. [11] utilized a local polynomial regression smoother for the removal of displacement artefacts in lung sounds. Resampling refers to modifying the sampling rate of the audio signals. Downsampling audio signals reduce the computational complexity and inference time for classification models without compromising their performance significantly. Segmentation algorithms identify the beginning and end of a respiratory cycle in the time domain. Spectrograms are two-dimensional audio representations helpful in extracting features using deep learning techniques such as CNN and its variants. Frequency ranges in which the audio signals have zero energy can be cropped from the spectrogram, allowing neural networks to focus on regions of interest thereby, leading to improved performance. Lastly, most neural networks often expect fixed length inputs and padding techniques such as zero-padding, duplicating respiration cycles are used to achieve a fixed-length input. Trimming eliminates any silence at the beginning or end of the audio.

Training machine and deep learning models on audio data requires a feature extraction step. This step transforms the raw audio into valuable features for each class that can be useful to the model in distinguishing between various classes. Audio features can be extracted from multiple domains such as time, frequency, time-frequency, entropy, spectral, statistical, wavelet, etc. [12] proposed a five-dimensional feature vector consisting of four time-domain features derived from Simple Moving Averages of a 92 ms window and one frequency domain feature (spectrum mean) to classify crackles. [13] proposed a feature vector consisting of averaged power-spectrums of 32 segments of a signal for classifying ten different samples of respiratory sounds. [14] decomposed the lung sounds into frequency sub-bands using wavelet transformed and extracted statistical features associated with each sub-band to represent the wavelet distribution. [15] used Mel Frequency Cepstral Coefficients that describes a signal's cepstrum for classifying lung sounds with Artificial Neural Networks. [16] constructed Autoregressive models using Linear Predictive Coding, a time-domain estimator of the signal based on a linear combination of previous samples weighted by LPC coefficients, for automated diagnosis of lung sounds. [17] used features such as percentile frequency ratios, mean crossing irregularity, kurtosis, Renyi entropy, etc. for the classification of wheeze and non-wheeze sounds. [18] proposed entropy-based features for the detection of four types of lung sounds and found the method to be robust against additive white Gaussian noise. [19] computed the "Wheeze Power Ratio", a ratio of the power of the maximum peak in the frequency range of 250–800 Hz to that of the mean power in the frequency range between 60–900 Hz and found it to be effective and robust in classifying wheeze sounds. [20] used audio spectral envelope and tonality index as discriminative features for wheezy sounds and found the method to be computationally simple, effective in detecting weak wheezes and robust against noise. The

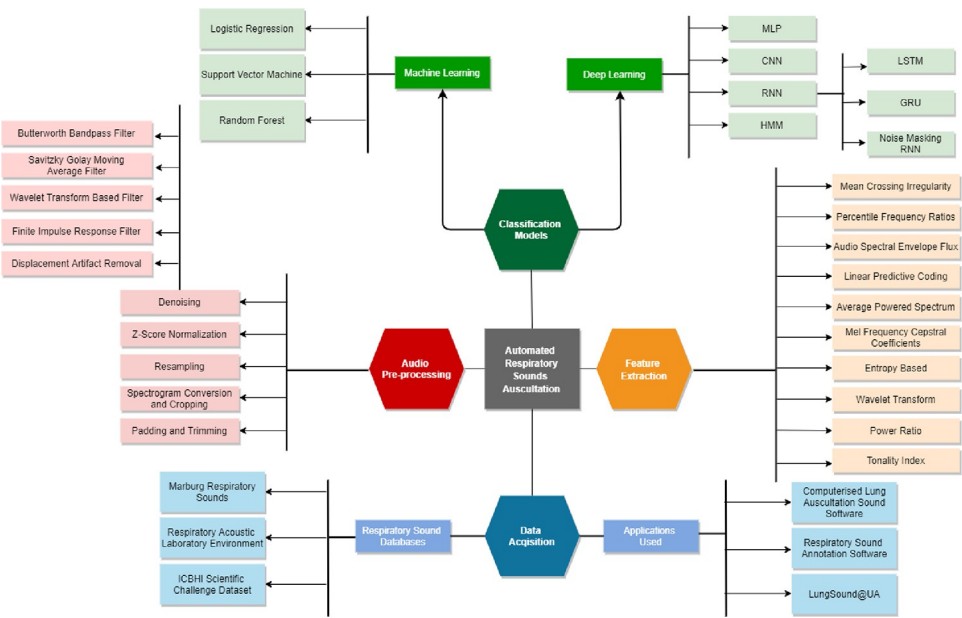

**Fig 1. Summary of techniques used in automated respiratory sounds auscultation.**

technique was suitable for automated auscultation of Asthma sounds acquired remotely from mobile devices where the wheezes may have weak intensity. Hence, deep-learning-based and manual feature extraction techniques have been proposed for classifying respiratory sounds.

Fig 1 and Table 1 collectively summarizes the commonly used pre-processing, feature extraction techniques, and classification models applied to lung sound signals. Table 1 categorizes the research works depending on the framework used i.e. either anomaly-driven (classifying respiratory cycles as crackles, wheezes, both or none) or pathology-driven classification (detection of specific diseases).

## 2.2 Literature review on audio data augmentation techniques

Data Augmentation is a technique that attempts to create new samples from existing samples with features that resemble the original samples in the training set. The generated samples vary slightly from the original training data and this helps a classifier to generalize better for certain classes. Data Augmentation techniques can be categorized into two types for audio data: the first is standard data augmentation techniques like Time stretching, pitch shifting, addition of white noise [28], signal speeding, frequency masking, time masking [29], etc and the second technique involves generating samples from generative models like Variational Autoencoders [30] and Generative Adversarial Network [31]. Table 2 summarizes the recent works undertaken towards augmenting audio datasets to enhance audio classification results.

## 3. Exploratory data analysis of ICBHI dataset

This section explores the various aspects of the respiratory diseases database provided by the International Conference on Biomedical Health Informatics. The database consists of 5.5 hours of annotated recordings with respiratory cycles containing crackles, wheezes or both [38]. Crackles are discontinuous sounds representing the presence of the fluid or secretions, whereas wheezing refers to high pitched whistling sound that occurs when a breath is taken [39, 40]. Wheezing is generally found in patients with Asthma and can be heard very loudly.

**Table 1. Literature review of classification models proposed for lung sound auscultation.**

| Author and Year | Framework | | Input/Features | Technique | Results |
|---|---|---|---|---|---|
| | **Pathology driven classification** | **Anomaly driven classification** | | | |
| [21] | ✗ | ✓ | Linear Predictive Cepstral Coefficients (LPCC) | Multilayer Perceptron Classifier | Accuracy of 99.22% was obtained |
| [22] | ✗ | ✓ | Mel Spectrograms with clipped black (zero energy) regions | RespireNet framework consisting of ResNet-34 trained on concatenation-based augmented samples of respiratory cycles along with device-specific optimizations. | Achieved a sensitivity of 0.54 and specificity of 0.83 in classifying wheezes (W), crackles (C), both wheezes and crackles (B) and healthy/normal (N) respiratory cycles. |
| [23] | ✗ | ✓ | Mel Frequency Cepstral Coefficients (MFCCs) and Power Spectrum Density (PSD) | For breath detector, models such as KNN, Random Forest and Logistic Regression were proposed. | All models achieved a precision of 0.98 and recall of 0.99, 0.98 and 0.99 respectively. |
| | | | | For anomaly detection engine, models such as Logistic Regression, SVM, ANN, Random Forest and KNN were used. | SVM and Logistic Regression achieved a precision and recall of 0.93, 0.94 and 0.91, 0.91 respectively, All other models achieved a precision and recall of 0.92 and 0.91 respectively. |
| [24] | ✗ | ✓ | Short Time Fourier Transformation (STFT) | Pretrained Deep Convolutional Network + SVM Classifier | Accuracy of 65.5% was obtained |
| | | | | Pretrained Deep Convolutional Network fined tuned for classification | Accuracy of 63.09% was obtained |
| [25] | ✓ | ✓ | Mel Frequency Cepstral Coefficients (MFCCs) | Recurrent Neural Networks like LSTM, GRU, BiGRU and BiLSTM | Sensitivity and Specificity of 64% and 82% were obtained respectively |
| [26] | ✗ | ✓ | Mel Frequency Cepstral Coefficients (MFCCs). | Noise Masking Recurrent Neural Network (NMRNN) | Sensitivity and Specificity of 56% and 73.6% were obtained respectively for end to end classification |
| [27] | ✗ | ✓ | Mel Frequency Cepstral Coefficients (MFCCs). | Hidden Markov Models in combination with Gaussian mixture models | Best Score achieved in second evaluation phase of ICHBI was 39.56 |
| [19] | ✓ | ✗ | Abnormal lung sounds, presence of breathlessness, peak meter readings and family history | Logistic Regression with L1 Regularization | Achieved 0.95 AUC score in separating COPD and asthma patients from other categories of diseases and 0.97 AUC score in distinguishing COPD and Asthma |

On the other hand, crackles are heard through a stethoscope and are generally a sign of fluid in the lungs. Early expiratory crackles occur in people with severe airway obstruction. These types of crackles mainly occur in COPD and Asthma. Crackles are further categorized as fine and coarse. Fine crackles occur in pulmonary edema, Pneumonia, and fibrosis cases, whereas coarse crackles occur in cases of Bronchiectasis [39]. Fig 2 shows the distribution of crackles and wheezes in respiratory cycles of various diseases.

The pie chart in Fig 3 shows the distribution of the patients with different respiratory diagnoses. The total number of patients who participated in the study was 126 from various age categories and gender.

The study produced 920 annotated audio samples belonging to various classes as shown in Fig 4.

Each audio file has more than one respiratory cycle of various lengths, which could be determined by the annotations provided in the dataset. The number of respiratory cycles for various diagnoses is shown in Fig 5.

The total number of respiratory cycles was 6898, of which 5641 cycles belonged to the COPD class. Fig 5 shows that the dataset is highly imbalanced. The volume of data for certain classes is negligible; hence there is a need to use data augmentation methods to ensure that the deep learning classifiers perform well.

**Table 2. A literature review of data augmentation techniques for audio classification.**

| Author and Year | Purpose | Data Augmentation Technique(s) | Input to augmentation technique | Results / Impact of augmentation on the performance of classification models |
|---|---|---|---|---|
| [28] | Environmental Sound classification | Time Stretching, Pitch Shifting, Dynamic Range Compression and Background Noise Addition | Log-Mel-Spectrogram | The accuracy for the proposed CNN (SB-CNN) increased from 73% (before augmentation) to 79% (after augmentation) |
| [32] | Speech Recognition | Mixup Augmentation | Normalized Spectrogram | The authors compared the classification performance of a VGG-11 model trained with empirical risk minimization and mixup augmentation and observed a lower classification error with mixup augmentation. |
| [33] | Speech Recognition | Variational Autoencoder | Discrete Fourier Transform | The authors proposed four classification models and evaluated these using Word Error Rate (WER). However, all four classification models suffered an increase in the WER after augmentation. |
| [29] | Speech Recognition | SpecAugment | Log Mel Spectrogram | Listen Attend Spell obtained WER of 2.8 with Augmentation and without presence of Language Model whereas LAS obtained WER of 4.1 without Augmentation |
| [34] | Acoustic Scene Classification | Spectrogram Rolling and mixup | Mel Frequency Cepstral Coefficient | The mean accuracy obtained by ResNet model before augmentation is 80.97% and after augmentation is 82.85% |
| [35] | Monaural Singing Voice Separation | Variational Autoencoder- Generative Adversarial Network (VAE-GAN) | Short Time Fourier Transform | The authors used metrics such as Source to Interference ratio (SIR), Source to Artifacts ratio (SAR) and Source to Distortion ratio (SDR) to evaluate the separation quality of a deep recurrent neural network and VAE-GAN. A higher value suggested better separation quality. The results revealed that VAE-GAN had a higher SDR and SAR whereas RNN had a higher SIR. |
| [31] | Environmental Sound Classification | WaveGAN | Raw audio | For baseline method the accuracy generated was 94.84% whereas after application of GAN the accuracy achieved was 97.03 |
| [36] | Animal Audio Classification | Signal Speed scaling, Pitch Shift, Volume increase/decrease, Addition of random noise and Time shift | Raw audio | The mean accuracy obtained by VGG19 on the cat dataset is 83.05 without augmentation and 85.59 with augmentation |
| | | pitch shift, time shift, summing two spectrograms from same class, applying random cropping followed by cutting the spectrogram in 10 different temporal slices and applying a function on it, application of time shift by randomly picking the shift T. | Mel Spectrogram | The mean accuracy obtained by VGG19 on cat dataset is 83.05 without augmentation and 90.68 with augmentation |
| [30] | Abnormal Respiratory Sounds Detection | Convolutional Variational Autoencoder | Mel Spectrogram | The specificity, sensitivity and F-score of the respiratory sounds classification model increased from 0.286 to 0.986, 0.888 to 0.988 and 0.349 to 0.900 upon augmentation respectively. |
| [37] | Acoustic Scene Classification | Zero-value Masking<br>Mini-batch based mixture masking<br>Mini-batch based cutting masking | Log Mel Spectrogram | The accuracy on DCASE 18 dataset is 76.2%<br>The accuracy on DCASE 18 dataset is 77.0%<br>The accuracy on DCASE 18 dataset is 76.9% |

Fig 6 shows the split of the real audio segments into train and test sets. The stacked bar chart reveals that nearly 70% of the audio segments in each class were used to train the VAEs and classification models. The remaining segments were reserved for testing/performance evaluation of the classifiers.

## 4. Research gaps

1. Existing research works [21–25] have shown that machine and deep learning techniques hold promise in automating respiratory sounds auscultation. However, developing a

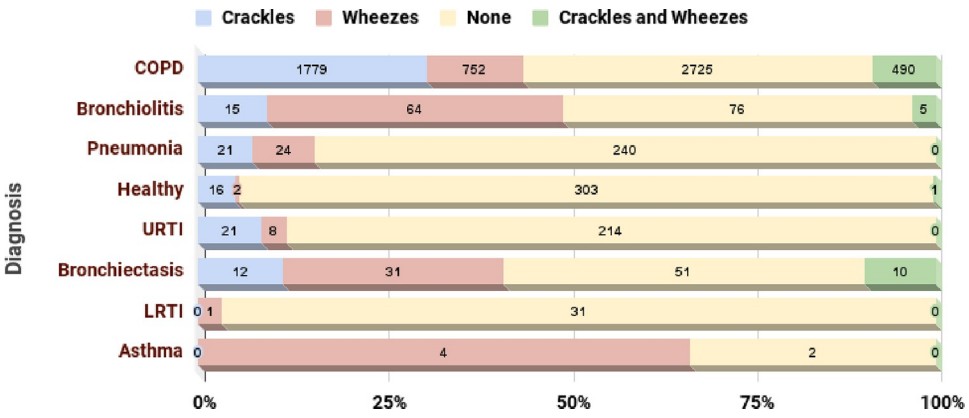

**Fig 2. Distribution of crackles and wheezes in the respiratory cycle.**

machine/deep learning model with good generalization ability, especially towards the minority/disease classes, requires a large volume of data to learn the desired characteristics of these classes. The total number of respiratory cycles for certain classes in the ICBHI data-set is negligible, making it impossible to train data-hungry deep learning models. Hence, a need for techniques that enable training deep neural networks with limited data has arisen. Data Augmentation is one such technique that can help deep learning models combat over-fitting in scenarios where certain classes are highly under-represented. We observed that very few works in the literature have experimented with the impact of data augmentation on the performance of respiratory sound classification models. Hence, this work has explored the efficacy of data augmentation using VAEs in enhancing lung sound classifica-tion performance.

2. Table 1 reveals that most works [21–24, 26, 27] on respiratory sounds classification have adopted an anomaly-detection based framework. Very few studies have focused on

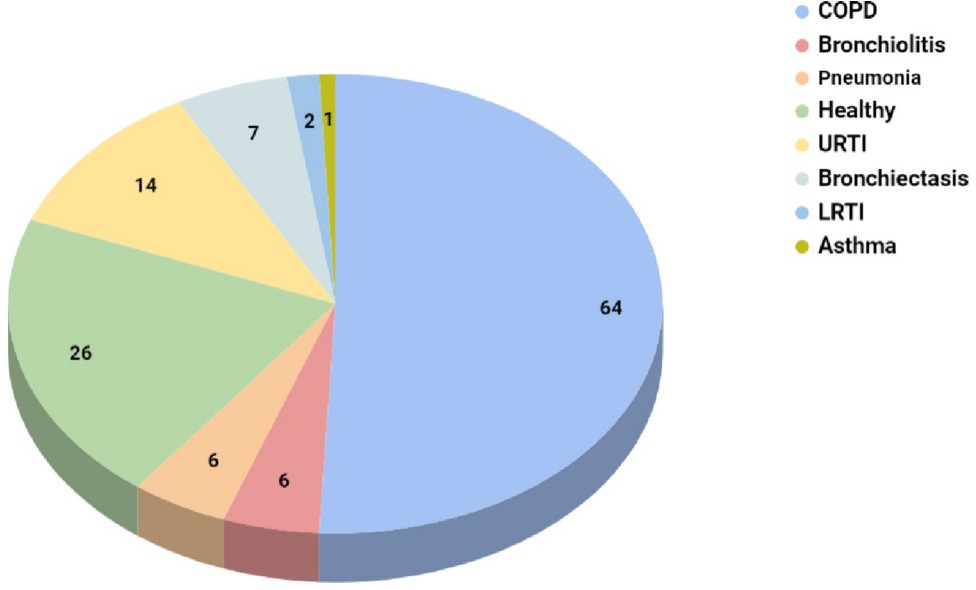

**Fig 3. Patient wise diagnosis in ICBHI dataset.**

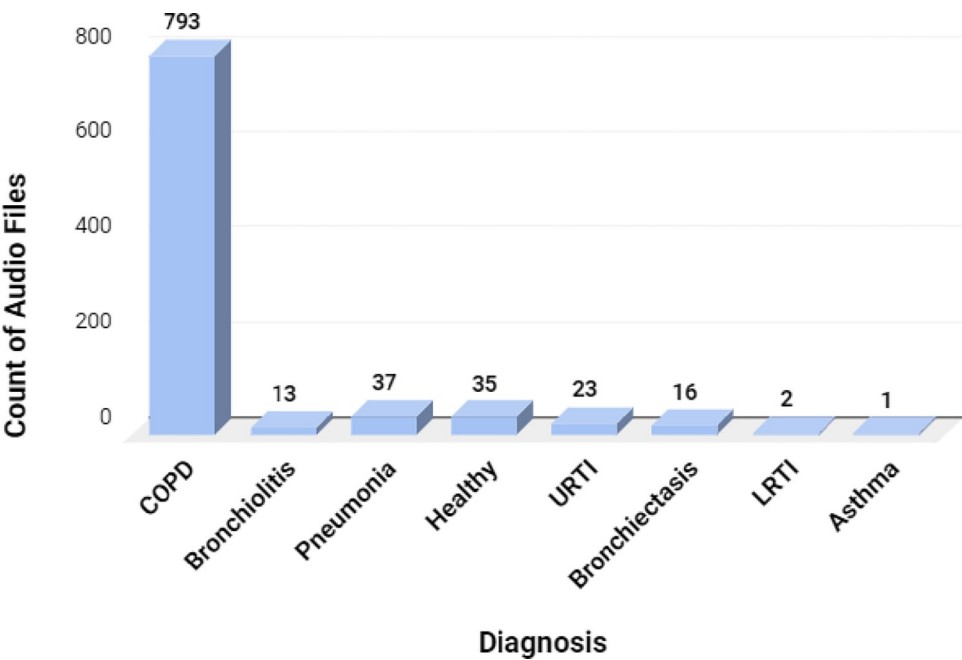

**Fig 4. Count of audio files for various respiratory diseases.**

classifying multiple respiratory diseases, probably due to insufficient labelled audio data. In this study, we aimed at achieving an acceptable classification performance metric for six disease classes present in the ICBHI dataset. Neural networks have the potential to learn sophisticated and robust feature representations of the disease classes. However, the small number of audio samples in the ICBHI dataset limits the creation of multi-class classification models. Hence, to leverage the power of deep learning for diagnosing multiple respiratory diseases, we need a large number of labelled samples for the disease classes. Creating a

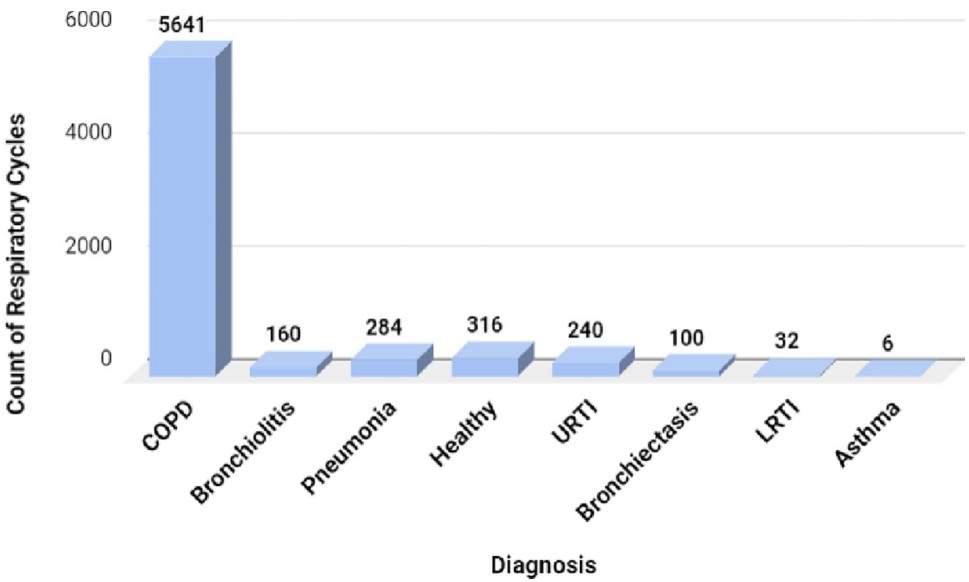

**Fig 5. Distribution of respiratory cycle per class.**

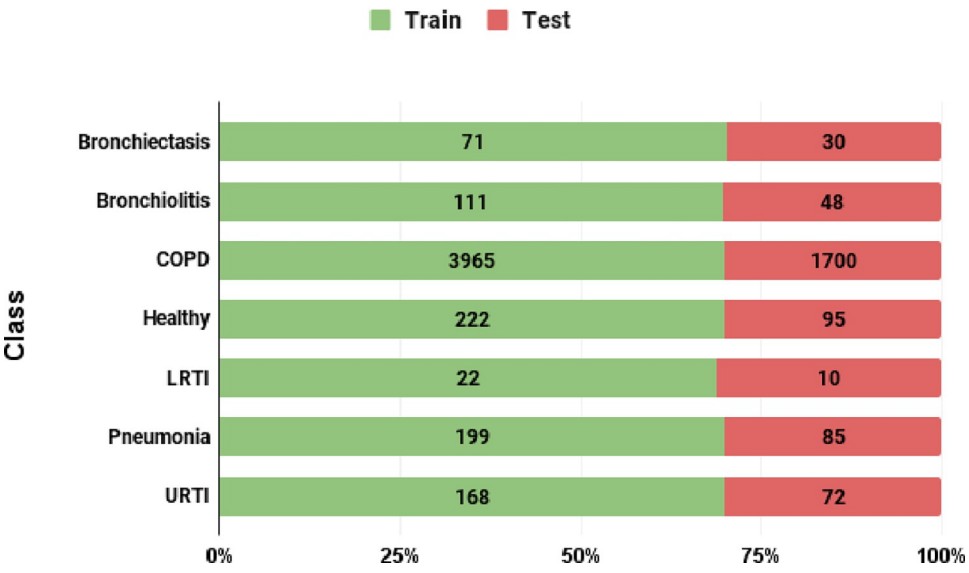

**Fig 6. Class wise split of audio segments into train and test sets.**

comprehensive database with a large volume of data for the minority classes is time-consuming, costly and patients belonging to these classes might not be available. Hence, generative modelling techniques for synthesizing samples of the minority/disease classes are required.

3. Standard audio augmentation techniques such as time-shifting, pitch shifting, noise addition, mixup, speed scaling, frequency masking, etc. can be used to create limited audio samples for the minority classes, both in quantity and variety, which may not be sufficient to significantly enhance the respiratory sound classification performance metrics or generalization ability. On the other hand, generative models such as VAEs and GANs that are less explored for audio synthesis in the healthcare domain can create a much wider set of augmentations that can lead to more robust classifiers. Lastly, we noticed that GANs were more prevalent in literature for audio generation than VAEs in various domains. However, training GANs is challenging due to instabilities in the training process and mode collapse [41]. Very few works [30] have reported generating biomedical signals such as respiratory sounds with VAEs. Moreover, we addressed three major issues that we noticed in [30]: 1. Inclusion of test set data for training the generative model (data leakage), 2. Lack of quantitative and qualitative assessment for the quality and variety of the synthetic samples and 3. Performance evaluation of the classifiers on the synthetic samples. These issues led to optimistic performance metrics on the test set, which could be misleading. Hence, our work provides an accurate insight into the potential of VAEs for augmentation and their role in enhancing classification performance metrics for respiratory sounds.

## 5. Methodology

Section 3 highlights the severity of the imbalance in the audio dataset. To address the issue, Variational Autoencoders were proposed to generate synthetic samples. Our work consists of two major steps: 1. generating synthetic samples for various respiratory classes and 2. Classifying the samples using deep learning models. As shown in Fig 7, after the audio acquisition, data pre-processing involves segmentation of the audio samples and padding the audio signals

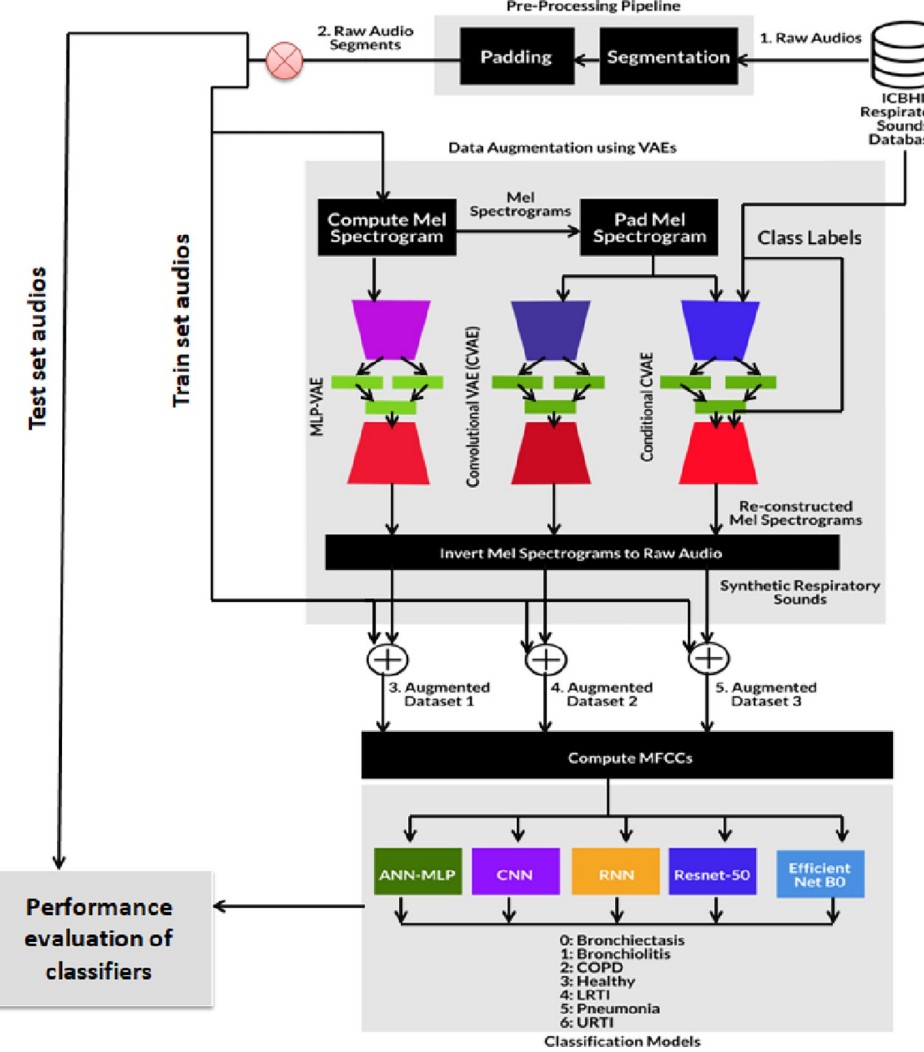

**Fig 7. Proposed methodology.**

to make them of the same duration. We aim to create a comparison in the performance of the classifiers before and after augmentation. For the classification of respiratory diseases, Mel Frequency Cepstrum Coefficients are obtained in feature extraction steps which are then passed to the proposed classification models. We propose three different variational autoencoders, namely Multilayer Perceptron VAE, Convolutional VAE and Conditional VAE. Unlike Generative Adversarial Networks, Variational Autoencoders do not take raw audio samples; hence the feature extraction step is necessary. Mel Spectrograms are considered for this step instead of Mel Frequency Cepstrum Coefficients because the former can be easily converted into audio samples.

## 6. Implementation

### 6.1. Audio acquisition

The dataset consists of audio samples which were obtained by two research teams over a different span and geographical location [38]. As indicated in Fig 2, the recordings can contain

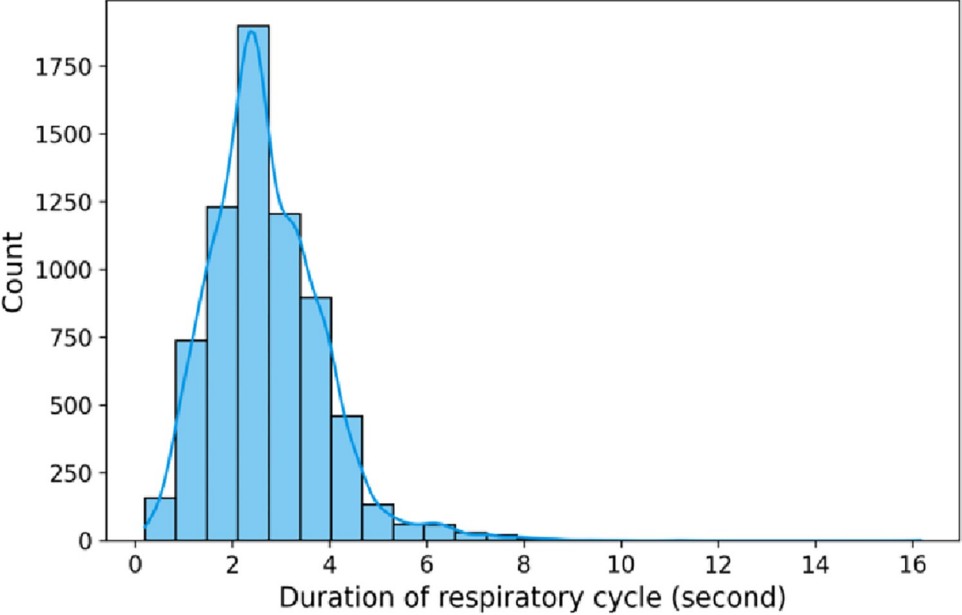

**Fig 8. Histogram showing the distribution of respiratory cycle durations.**

crackles, wheezes or both. The dataset contained 1864 recordings with crackles, 886 with wheezes, and 506 with both wheezes and crackles. The dataset contained various respiratory diseases obtained from 126 subjects, as mentioned in Figs 3 and 4. Sound annotation for the audio samples was done by two physiotherapists and one medical doctor with experience in visual-auditory wheezes/crackles identification and with the help of Respiratory Sound Annotation Software. The result of the annotation stage was the generation of a text file for each audio recording which was then used in the pre-processing step.

## 6.2 Pre-processing

**6.2.1 Segmentation.** The dataset consists of audio recordings of varying length i.e. 10 to 90 seconds. Each audio sample is broken down into respiratory cycles or segments depending on the annotation provided. Fig 8 describes the distribution of the segments based on its length.

**6.2.2 Padding.** As shown in Fig 8, the histogram peaks at 2.5 seconds and the majority of the respiratory cycles i.e. 6779 respiratory cycles lie before 6 seconds duration. We discarded those respiratory cycles which were above the length of 6 seconds and padded the respiratory cycles less than 6 seconds with silence in the end to ensure that all the segments were of the same length. The padded audio waveforms for different respiratory classes are shown in Fig 9.

## 6.3 Data augmentation

Variational Autoencoders belong to the family of Generative Models. Variational Autoencoders bear a huge similarity to Autoencoders in terms of their design. They consist of an encoder (a recognition or inference model) and a decoder, also known as a generative model. Another similarity between the two is they attempt to reconstruct the input data while learning from the latent vectors. However, the difference between the two is that the latent space of VAE is continuous. This is achieved as the encoder does not output an encoding vector of

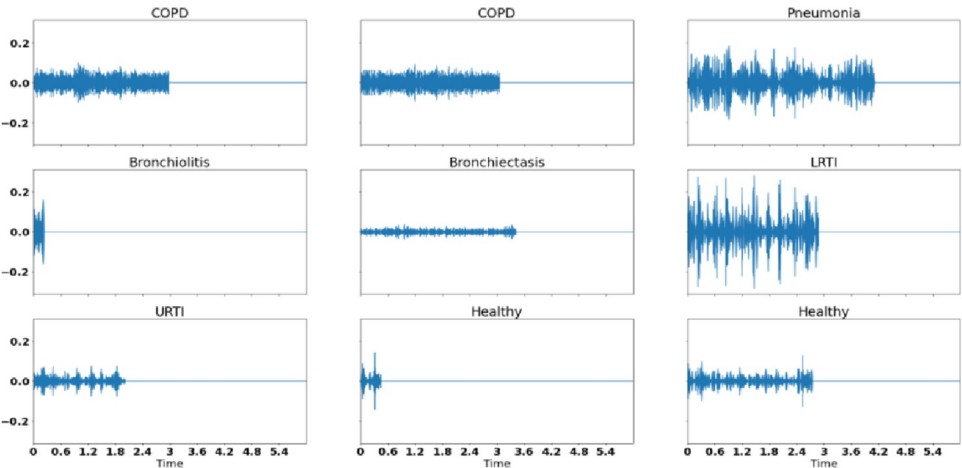

**Fig 9. Padded raw audio segments of all classes used in the study.**

length n but two different vectors of length n, one, a vector of means and another vector of standard deviation, as shown in Fig 10.

The purpose of the mean vector is to control the center location of the encoded input whereas the purpose of the standard deviation vector is to control the area in which the encoding can vary. Since the encoding is generated randomly in the circle, the decoder also exposes itself to variations of encoding.

Since the proposed Variational Autoencoders do not take raw audio as input, it is essential to carry out a feature extraction step. In this paper, we use Mel Spectrogram as a feature extraction step before the data augmentation process takes place. Mel spectrogram is a spectrogram where the frequencies are converted into the Mel scale.

Let $\{X = x_i\}_{i=1}^{N}$ be the training data where each $x_i \in \mathfrak{R}^d$ represents a mel spectrogram belonging to a given respiratory class. The encoder in the VAE learns a mapping $Q_\theta(z|X)$ from an input $x_i$ to the mean $\mu(x_i)$ and covariance $\sigma^2(x_i)$ vectors of the latent variables. VAEs assume

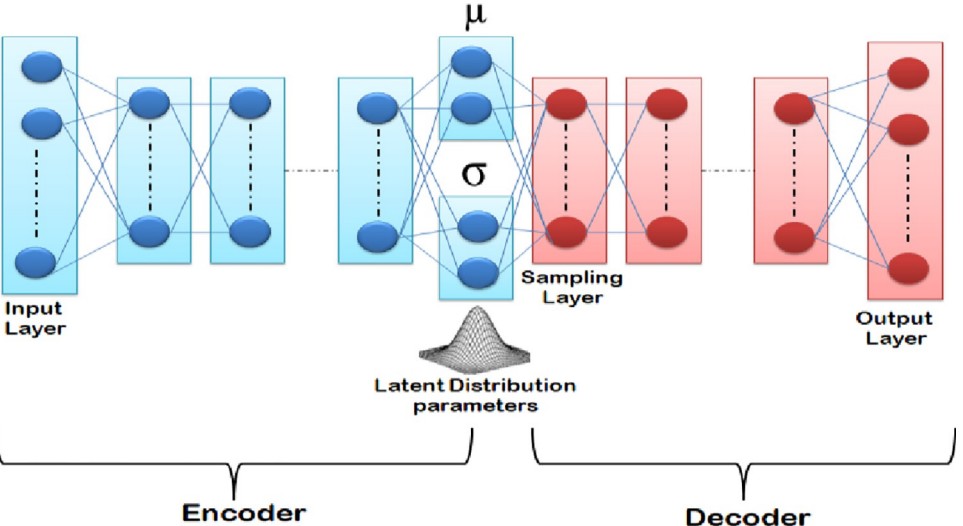

**Fig 10. Structure of variational autoencoder.**

that the latent variable z follows a standard normal distribution $N(0, 1)$. The decoder needs to sample from the latent distribution outputted by the encoder, since the encoder no longer outputs a latent representation z as in autoencoders. The decoder of a VAE learns a mapping $P_\phi(X|z)$ from the latent representation z to the distribution parameters of the training data X. For the decoder to generate realistic samples, it needs to maximize the log likelihood of the input data i.e. log (p(X)). We also need to ensure that the VAE does not memorize/overfit the training data. This is done by adding a regularization term to the loss function of the VAE that constrains the Gaussian distribution outputted by the encoder to be close to a standard normal distribution. A standard normal distribution has an identity covariance matrix which implies that the covariance between the latent variables is zero and they are independent of each other. Hence, a VAE which learns to model the training data distribution p(X) with an encoder $Q_\theta(z|X)$, a decoder $P_\phi(X|z)$ and a latent distribution p(z) is trained using the objective function (1)

$$max_{\theta,\phi} log(p(X)) \approx max_{\theta,\phi} \mathrm{E}_{Q_\theta(z|X)}[log(P_\phi(X|z))] - D_{KL}(Q_\theta(z|X)||p(z)) \tag{1}$$

where $\theta$, $\phi$ are the parameters of the encoder and decoder respectively and $D_{KL}$ represents the Kullback-Leibler Divergence between two probability distributions.

The following steps obtain the Mel spectrograms, which are fed to the variational autoencoder:

1. Sampling the input with the window size of n_fft as 2048 while making hops of 512 to sample the next window.

2. Computation of Fast Fourier Transform for each window to transform the time domain to frequency domain

3. Generation of Mel Scale is done by taking the entire spectrum and breaking it into 128 bins.

4. Generation of spectrograms.

The Mel spectrograms generated by the above process are shown in Fig 11.

We propose three variants of Variational Autoencoders for tackling data imbalance. These are Multilayer Perceptron VAE (MLP-VAE), Convolutional VAE (CVAE) and Conditional CVAE. The architectural details, along with hyperparameter settings, are discussed in the following subsections. All the VAE models were trained on mel spectrograms of the minority classes. The encoders of the VAEs aimed at learning a mapping from the high-dimensional mel spectrograms to low-dimensional latent representations, which in turn can be used by the decoder to reconstruct a realistic mel spectrogram for a given minority class. As highlighted in Eq (1), the decoder learns to generate mel spectrograms similar to the real spectrograms by minimizing the reconstruction loss (Mean Squareddd Error in our case). On the other hand, the encoder is incentivized to output normally distributed latent variables by the regularization term (KL divergence between the distribution of latent variables and a standard normal distribution in our case). To control the outputs generated by the unconditional decoders, we instantiated a new VAE model per minority class. Each instance of the unconditional models was trained on only one of the minority classes, thus ensuring that all synthetic mel spectrograms generated by an unconditional model belong to the same class only. On the other hand, the Conditional VAE model was trained on all six minority classes simultaneously. Upon completing training for an instance of a VAE model, we generated synthetic mel spectrograms using the decoder network by inputting a latent vector z sampled from a standard normal distribution and a class label for the Conditional VAE.

**6.3.1 Multilayer perceptron VAE.** The encoder consists of three dense layers, with the third layer generating the mean and the log variance. The addition of the log variance layer

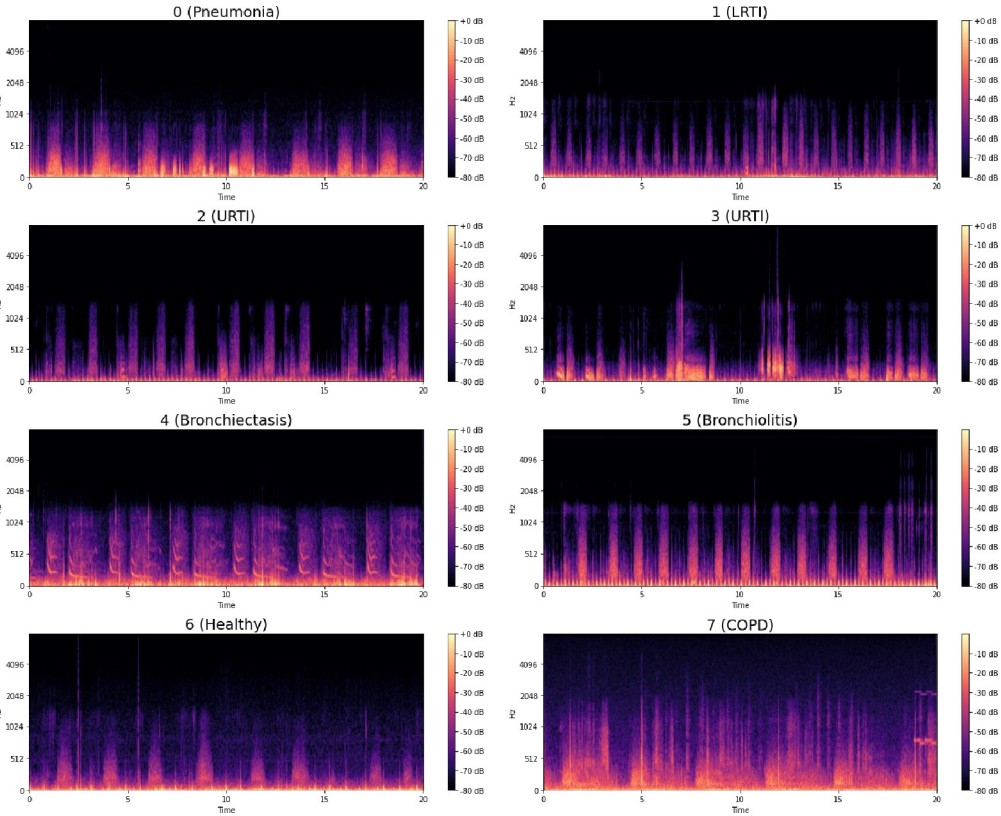

**Fig 11. Mel spectrograms of various respiratory diseases.**

makes the computation of KL loss and re-parameterization easier. The decoder also consists of two intermediate dense layers and an output layer that takes samples of z to reconstruct a Mel spectrogram. The dimension used in the intermediate layer is 512. The size of the latent dimension is 2. The lambda function implements the re-parameterization trick to push the sampling stochastic latent codes outside the network. The optimizer used was Adam. The activation function used in the intermediate and the output layers are ReLU and sigmoid, respectively. The network configuration of the multilayer perceptron VAE is shown in Fig 12.

**6.3.2 Convolutional VAE.** The encoder is made up of two convolutional layers and three intermediate dense layers to generate the latent code. The kernel size used in the convolutional layers is three, numbers of strides were assigned as two and filters were varied depending on the number of convolutional layers i.e. there were 16 for the first layer and 32 for the second layer. The decoder has one dense layer, a reshape layer which is then given as an input to the three Conv2DTranspose layers responsible for upsampling and reconstructing the image to its original dimension. The filters in Conv2DTranspose layers vary with the layers and are assigned 32, 16 and 1, respectively. The strides and kernel size for the Conv2DTranspose were assigned 2 and 3, respectively. The activation function assigned to the first two layers was Rectified Linear Activation function or ReLU and sigmoid activation function for the last upsampling layer. The optimizer used in this case was RMS Prop. The network configuration of the VAE is indicated in Fig 13.

**6.3.3 Conditional VAE.** If the latent space is randomly sampled, it becomes impossible for the Variational Autoencoder to control the class of the sample being generated.

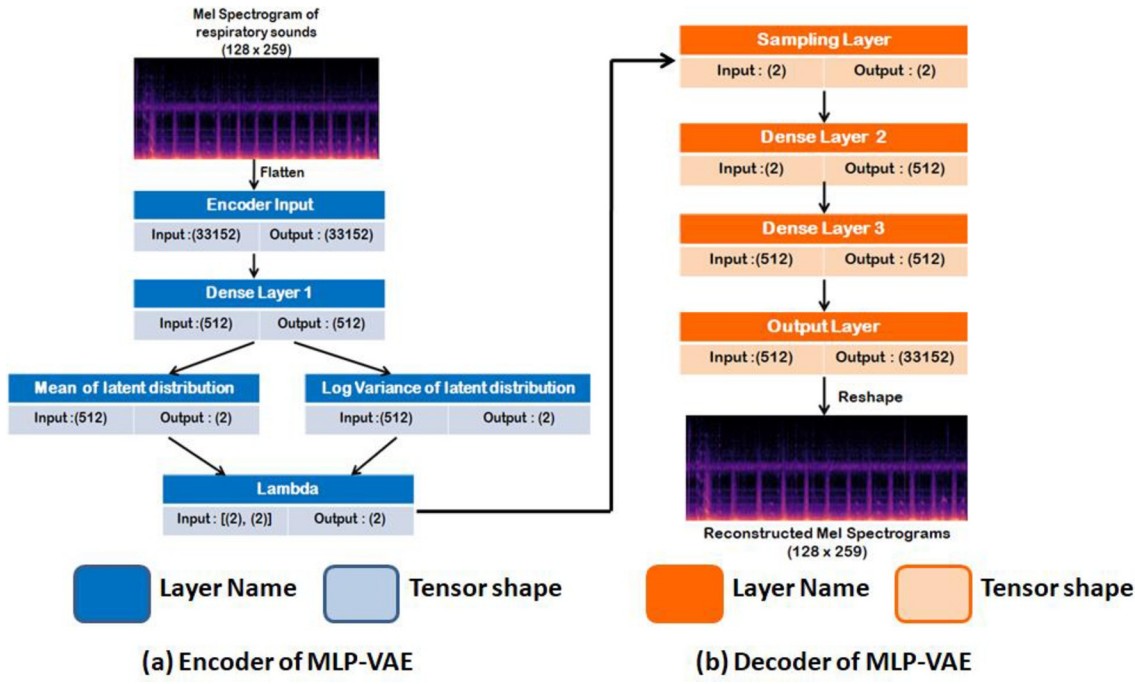

**Fig 12. Overall architecture of MLP-VAE.**

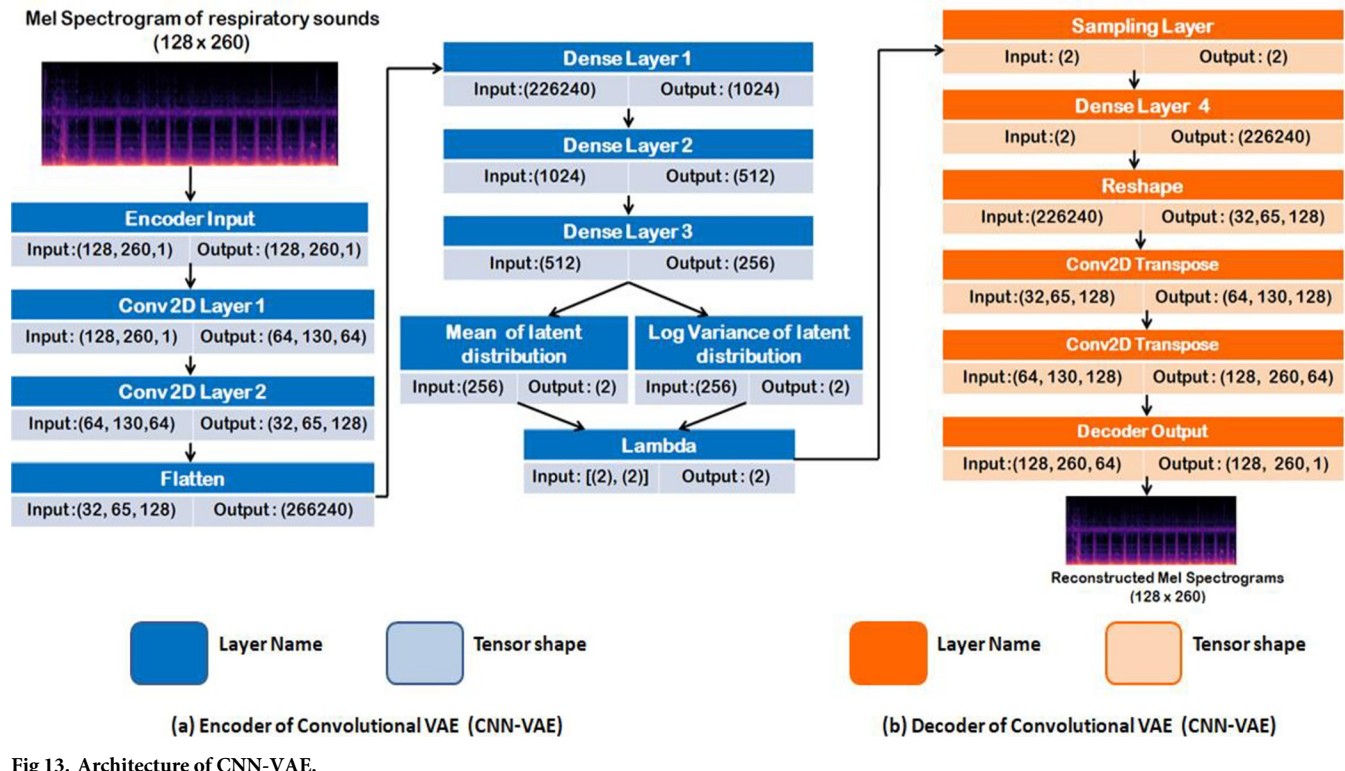

**Fig 13. Architecture of CNN-VAE.**

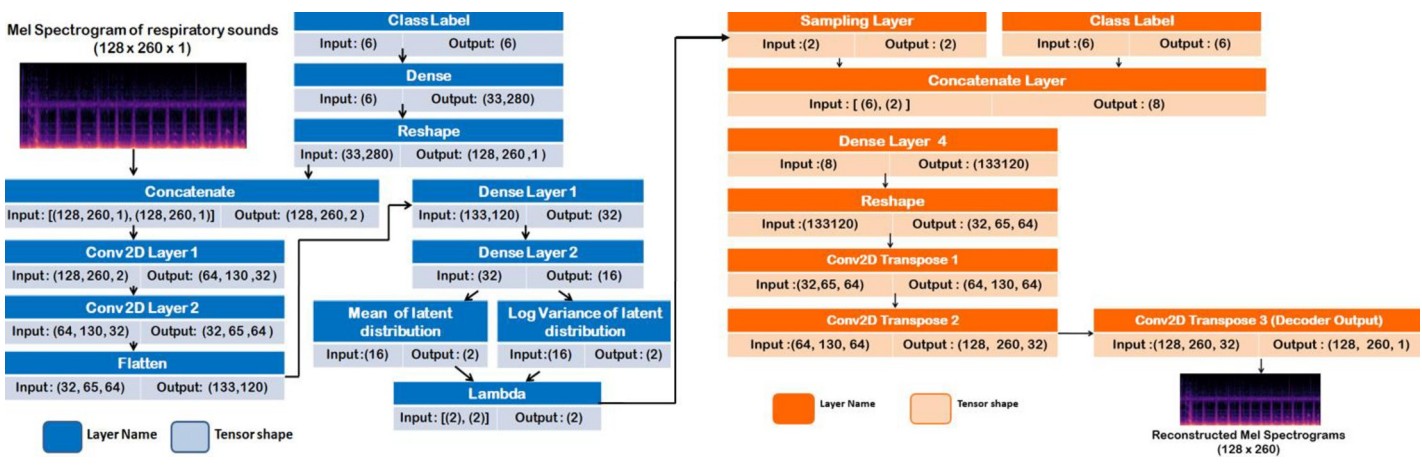

**Fig 14. Overall architecture of conditional VAE.**

Conditional VAE addresses the issue by including a one hot label or a condition into the encoder and the decoder. The reconstruction loss of the decoder and KL loss of the encoder is determined by latent vector and the condition. Conditional VAE is a special type of β-VAE (Disentangled VAE) where β = 1. The network structure of the VAE consists of an encoder which has a one hot vector of the labels provided as the input along with 3 Dense layers, 2 Convolutional layers with varying filters i.e. 16 and 32, kernel size as 3 and latent dimension as 2. The structure of the decoder consists of one hot vector of the class labels and 2 dimension input obtained from the lambda function as the input. It consists of 1 Dense layer and 3 Conv2DTranspose Layers with filters 32, 16 and 1 respectively. The optimizer used in this case was RMS Prop. The network configuration of the VAE is indicated in Fig 14.

Variational Autoencoder generates reconstructed Mel spectrograms which are converted into audio by applying Griffin-Lim [42]. The numbers of audio samples generated for the minority classes by various variational autoencoders are shown in Table 3.

## 6.4 Feature extraction for classification models

Before feeding the data into neural networks for the classification of respiratory diseases, it is essential to extract the required features from the data as it reduces the computation time of the classification models. We propose the use of Mel Frequency Cepstrum for the extraction of features from audio. This technique involves windowing the signal, applying direct Fourier transformation, taking log of magnitude and warping the frequencies on the Mel scale, and finally applying inverse Direct Cosine Transformation. Fig 15 shows the steps of computing the Mel Frequency Cepstral Coefficients for an audio signal.

We gave the value of the number of MFCC as 13 and sample rate of each audio file as 22050. Fig 16 visualizes the MFCCs over time for various respiratory classes in the dataset.

**Table 3. Samples generated by proposed variational autoencoders.**

| Synthetic samples generated | LRTI | URTI | Bronchiectasis | Bronchiolitis | Healthy | Pneumonia |
|---|---|---|---|---|---|---|
| MLP-VAE | 5641 | 5641 | 5642 | 5641 | 5636 | 5637 |
| CNN-VAE | 5633 | 5640 | 5641 | 5640 | 5641 | 5640 |
| CVAE | 5657 | 5569 | 5577 | 5636 | 5646 | 5613 |

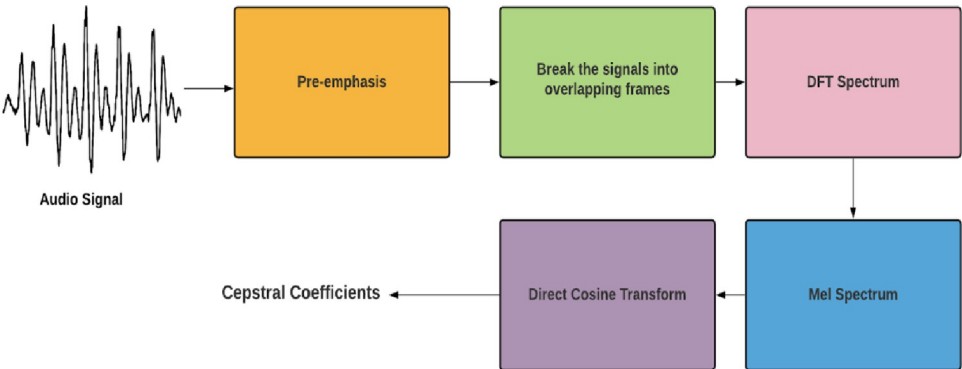

**Fig 15. Procedure for computing MFCCs.**

## 6.5 Classification models

In this paper, we have proposed five deep learning architectures for the classification of various respiratory diseases. The architectural details and hyperparameter settings of these models are discussed in the following subsections and Table 4, respectively. The last layer of these models is a dense layer with seven neurons and the softmax activation function that outputs the class probabilities corresponding to the seven respiratory classes. The classification models were compiled with categorical cross-entropy loss function, Adam optimizer and a learning rate of 0.0001. The classification models were trained on the imbalanced and augmented training sets and evaluated using metrics such as specificity, sensitivity, precision and F1-score.

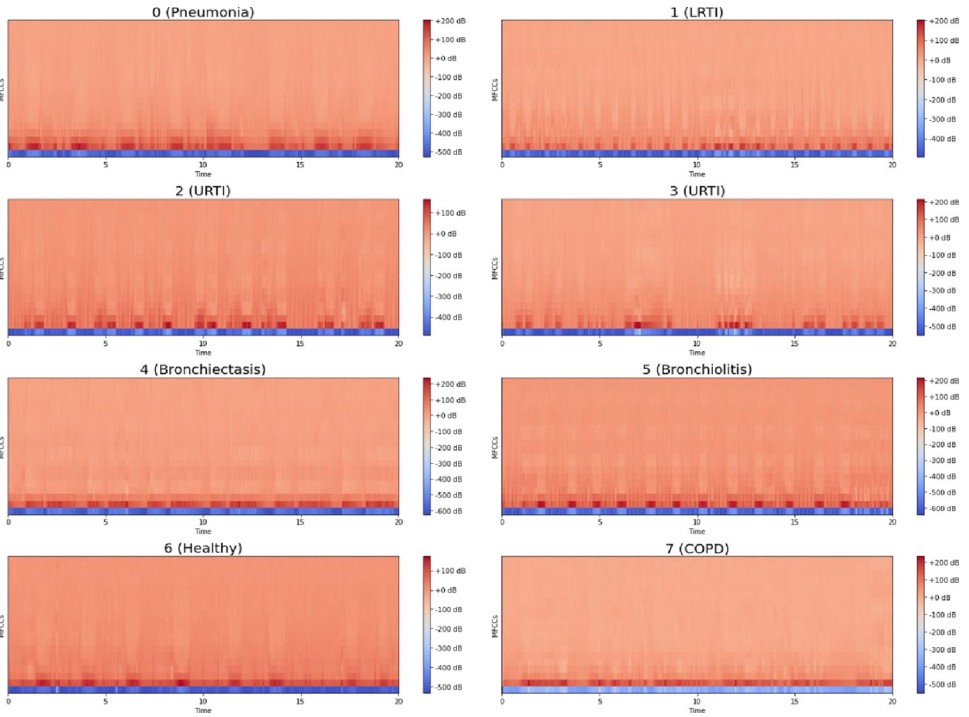

**Fig 16. MFCC for various respiratory classes.**

**Table 4. Hyperparameters configuration of the proposed classification models.**

| Model | Hyperparameters | Value |
|---|---|---|
| **Multilayer Perceptron** | Number of neurons in hidden layers (1–3), layer 4 and hidden layers (5–7) | 512, 1024, 512 |
| | Activation function used in hidden layers | ReLU |
| | Optimizer and learning rate | Adam and 0.0001 |
| **Convolutional Neural Network** | Number of filters in Conv2D layers | 32 |
| | Stride in Conv2D layers | (1,1) |
| | Pool size in MaxPool2D layers | (2,2) |
| | Stride in MaxPool 2D layers | (2,2) |
| | Kernel size in Conv2D layers 1 and 2 | (3,3) and (2,2) |
| | Number of neurons in Dense layers (1–4) | 64, 128, 128, 64 |
| | Activation function used in Dense layers | ReLU |
| | Optimizer and Learning Rate | Adam and 0.0001 |
| **LSTM** | Number of memory cells in LSTM layers 1 and 2 | 64 and 128 |
| | Number of neurons in Dense layers (1–3) | 64, 256, 128 |
| | Activation function used in LSTM layers 1 and 2 | tanh |
| | Activation function used in Dense layers (1–3) | ReLU |
| | Optimizer and Learning Rate | Adam and 0.0001 |
| **ResNet-50** | Number of neurons in Dense layers (1–6) | 256, 128, 64, 512, 512, 512 |
| | Activation function used in Dense layers (1–6) | ReLU |
| | Optimizer and Learning Rate | Adam and 0.0001 |
| **Efficient Net B0** | Number of neurons in Dense layers (1–3) | 256, 128, 64 |
| | Activation function used in Dense layers (1–3) | ReLU |
| | Optimizer and Learning Rate | Adam and 0.0001 |

**6.5.1 Multilayer perceptron.** As shown in Fig 17, the model takes flattened MFFC as input which is then fed to the seven dense layers which have dropout in the range of 10% to 40% in order to reduce overfitting.

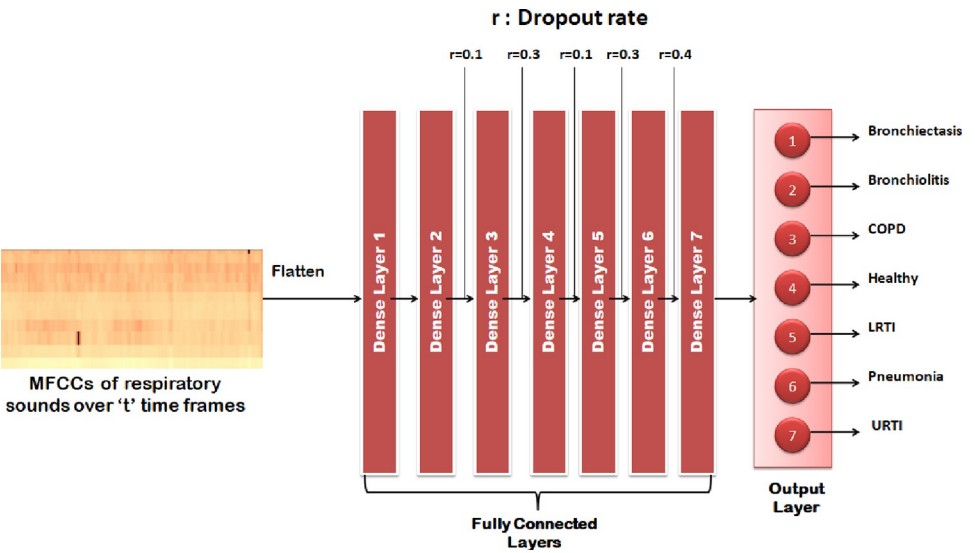

**Fig 17. Visual representation of MLP model.**

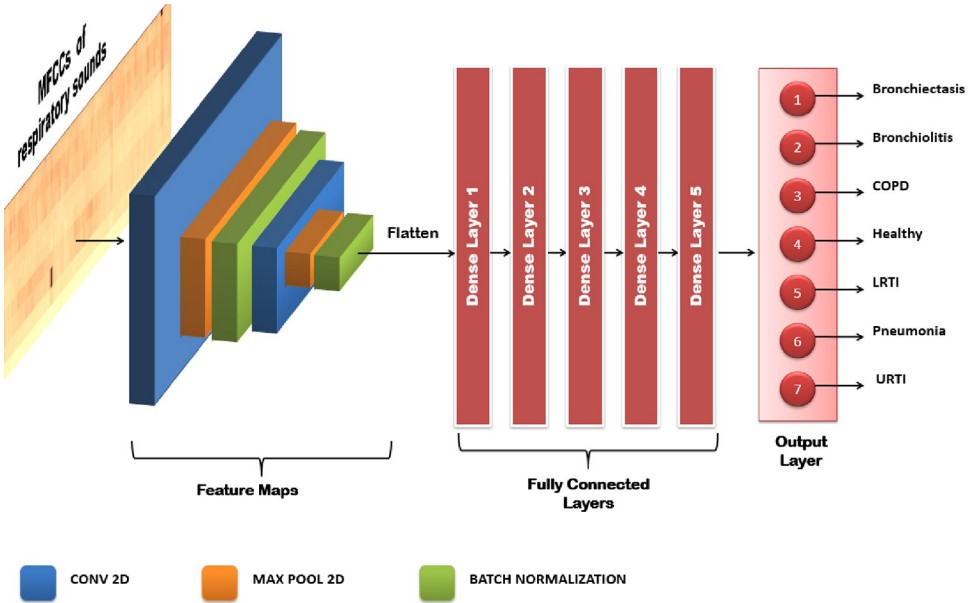

**Fig 18. Visual representation of CNN model.**

**6.5.2 Convolutional neural network.** The second model used for detection of respiratory diseases is CNN which has a sequential framework. The model has two Conv2D layers which were used for feature extraction, with five intermediate dense layers and a dense layer as the output layer. Convolutional Neural Networks have the ability to capture the spatial and temporal dependencies from the input, specifically an image through the application of various filters. It is preferred because of the reduction in the number of parameters involved and option of reusability of the weights. In short, Convolutional Neural networks reduce the mel-spectrograms, into a form that is easier to process and does not lose the critical features on its way to achieve it. Fig 18 shows the architecture of the proposed CNN model for classifying MFCCs of respiratory sounds.

**6.5.3 Recurrent neural network: LSTM.** A recurrent neural network is a class of neural networks where the previous outputs are used as input while having hidden states. The advantage of recurrent neural networks is that the weights are shared across time. On the other hand, vanishing gradients are one of the major problems in RNN. Hence, LSTM was introduced to counter the issue. LSTM has the edge over feed-forward and recurrent neural networks because of their property of selectively memorizing patterns for long durations of time. Fig 19 shows the architecture of the proposed LSTM model for classifying the MFCCs of respiratory sounds over several time frames.

**6.5.4 Resnet-50.** The proposed respiratory sound classification method involves a pretrained Resnet-50 model [43], which is considered for both deep feature extraction and fine-tuning. The input to the model is the MFCC spectrogram of dimension (13,130,1). The fully connected layers were used for predicting the class labels. The output obtained from Resnet is flattened, normalized using BatchNormalization and passed into fully connected layers. The fully connected layer consists of 6 hidden dense layers and one output layer with a softmax activation function. The dense layers have dropouts in the range of 10% to 30% and a batch normalization layer to prevent overfitting. Fig 20 shows the architecture of the proposed transfer learning model that employs the RESNET-50 backbone for extracting features from the MFCCs of respiratory sounds and classifies the same.

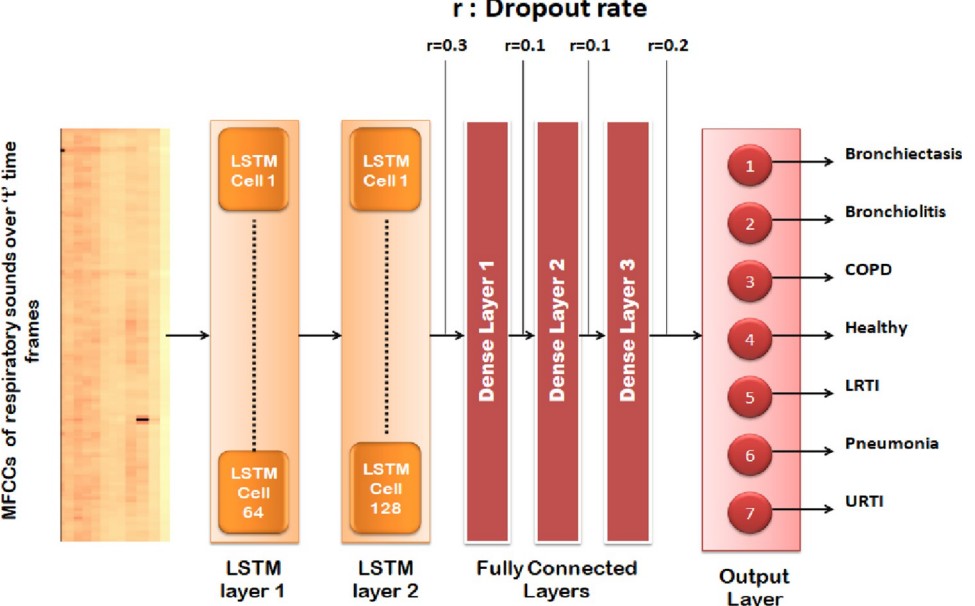

**Fig 19. Visual representation of RNN-LSTM model.**

**6.5.5 Efficient Net B0.** This model contains Efficient Net B0 [44] as the base model over which other layers are built. The input is fed into the Efficient Net Architecture and the output obtained from it is flattened. The input is of the size (13, 130, 1) and the dimension obtained after passing it through an efficient net B0 layer and flattening is 6400. The training set is normalized using BatchNormalization and dropout layers of 50% are added to reduce overfitting. There are 3 hidden layers with 256, 128 and 64 neurons respectively and one output layer with

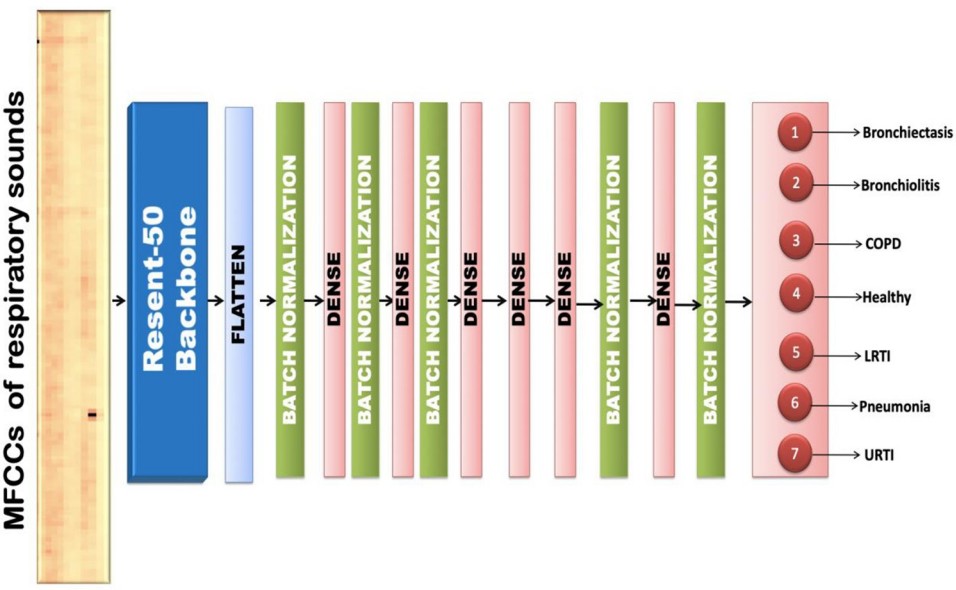

**Fig 20. Visual representation of RESNET-50 transfer learning model.**

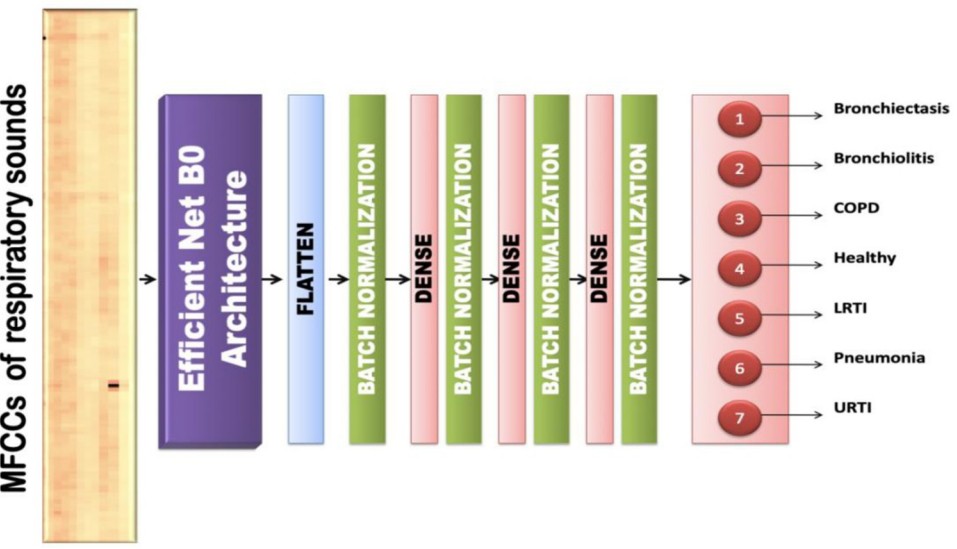

**Fig 21. Visual representation of EFFICIENT NET B0 transfer learning model.**

softmax as the activation function. The optimizer used was Adam with a learning rate of 0.0001. Fig 21 shows the architecture of the proposed transfer learning model that employs the Efficient Net B0 base architecture for extracting features from the MFCCs of respiratory sounds and classifies the same.

# 7. Results

In this section, we evaluate the performances of the generative and classification models. To assess the ability of the generative models to synthesize realistic respiratory sounds, we used metrics such as Fréchet Audio Distance (FAD), Cross-Correlation, Mel Cepstral Distortion and compared the features of the synthetic and original audios using Principal Component Analysis (PCA). To evaluate the performance of the classifiers and the effectiveness of the addition of the synthetic samples to the imbalanced training set in combating overfitting for the minority classes, we measured metrics such as confusion matrices, specificity, sensitivity, precision and F1 score on the hold-out test set.

## 7.1 Generative models

**7.1.1 Fréchet audio distance.** The Fréchet Audio Distance (FAD) is a reference-free evaluation metric used for assessing the quality of synthetic samples created by a generative model [45]. Reference samples for each synthetic audio generated are not required for measuring the FAD metric. Other benefits of using the FAD metric include robustness against noise, computational efficiency, correlates well with the human judgment of audio quality (Pearson Correlation: 0.52), sensitivity to intra-class mode dropping [46]. The FAD metric compares the statistics of embeddings obtained from a VGGish audio classification model for the original and synthetic datasets using Eq 2.

$$FAD = ||\mu_r - \mu_g||^2 + trace(\sum{}_r + \mu_g - 2\sqrt{\sum{}_r\sum{}_g}) \qquad (2)$$

where $\mu_r$: Mean of real data distribution
$\mu_g$: Mean of generated data distribution

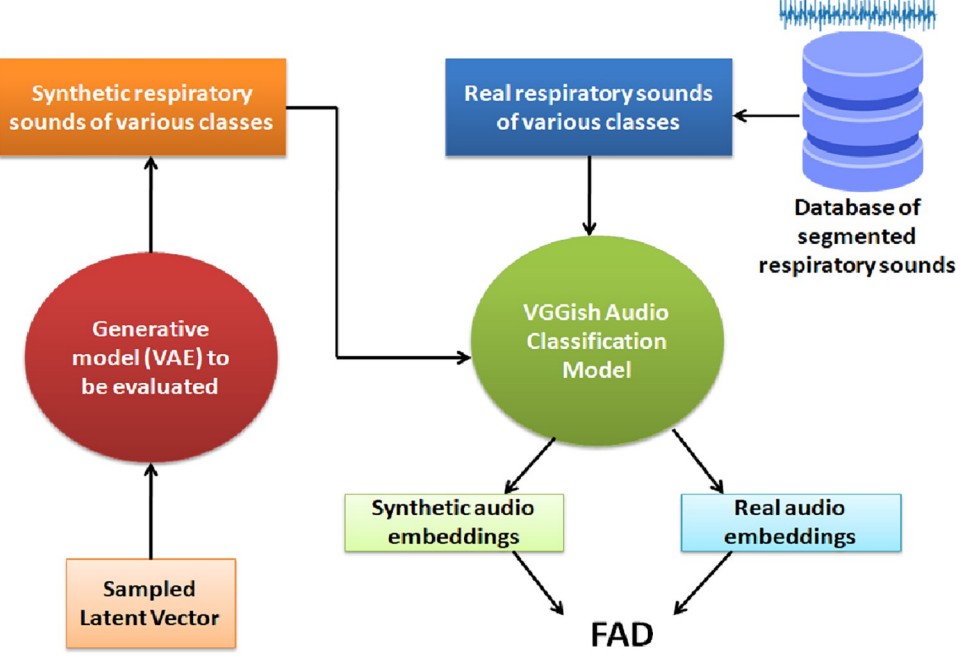

**Fig 22. Computation of FAD.**

$\sum_r$: Covariance of real data distribution

$\sum_g$: Covariance of generated data distribution

Fig 22 illustrates the computation of FAD. The synthetic and original audios for each lung sound class are inputs to the VGGish model to obtain the corresponding embeddings. The mean and covariance of these embeddings are obtained to compute the FAD metric. Lower the FAD, the smaller the distance between the distributions of the original and synthetic audios. A small FAD is also indicative of the good audio quality of the synthetic samples.

Table 5 and the bar graph in Fig 23 show the FAD scores of the synthetic audios generated by the proposed VAEs w.r.t the original sounds for each class. The FAD scores of synthetic audios created by CNN-VAE and Conditional VAE exhibit consistent values in the range of 10–14, whereas those of MLP-VAE vary greatly depending on the lung sound class. The MLP-VAE synthesized audios of the 'Healthy' and 'LRTI' classes with low FAD scores of 4.81 and 3.16, respectively. The synthetic audios are obtained by inverting the Mel spectrograms reconstructed by the decoder of the VAEs. Mel Spectrograms are a lossy compression of STFT as the phase information gets discarded. Hence, to estimate the phase during inversion, we used the iterative Griffin and Lim method [42] offered by the audio processing library Librosa. Hence, distortions introduced by the inversion method may lead to a higher FAD score.

**7.1.2 Visualization of audio features using principal component analysis.** The features of the real and synthetic respiratory segments were visualized in two-dimensional space using

**Table 5. FAD of synthetic samples of minority classes w.r.t real samples.**

| Generative model | Class | | | | | |
|---|---|---|---|---|---|---|
| | **Bronchiectasis** | **Bronchiolitis** | **Healthy** | **Pneumonia** | **LRTI** | **URTI** |
| **MLP-VAE** | 28.40 | 11.72 | 4.81 | 12.34 | 3.16 | 14.11 |
| **CNN-VAE** | 12.47 | 10.86 | 12.05 | 11.56 | 12.10 | 10.44 |
| **Conditional VAE** | 13.96 | 10.88 | 12.07 | 11.62 | 10.79 | 10.57 |

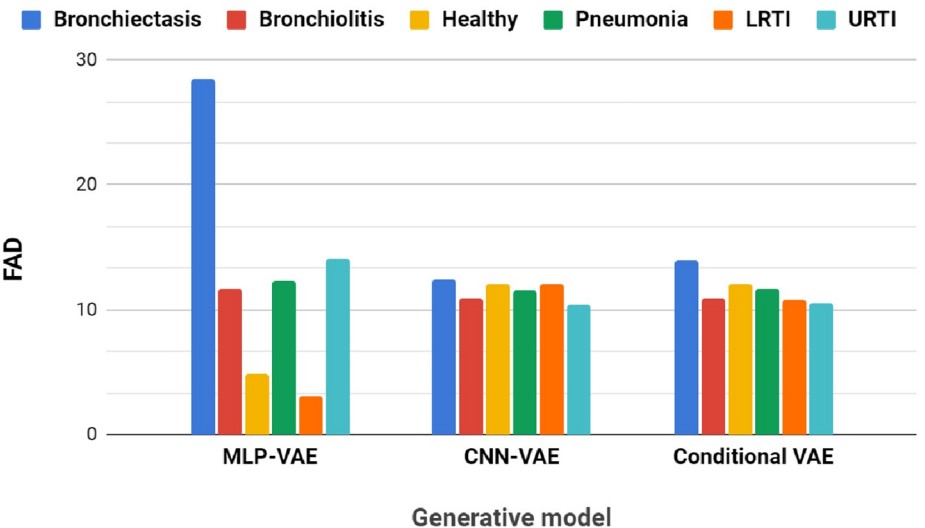

**Fig 23. FAD of synthetic samples w.r.t real samples for minority classes.**

Principal Component Analysis. For each VAE and lung sound class, 200 synthetic audios were sampled, whereas all respiratory cycles of the original audios were considered. The features extracted from these audios included the mean, standard deviation and the first-order derivative of their Mel Frequency Cepstral Coefficients. The feature extraction process resulted in a feature vector of 39 dimensions compressed to 2 using PCA. The scatter plots in Figs 24–26 show the spread of the two principal components for the synthetic and original audios.

The distribution of the features of the synthetic segments created by MLP-VAE shown in Fig 24 is linear and inconsistent with that of the original segments. The only exception is the "LRTI" class where the distribution of features of the synthetic samples follows that of the real audios closely. The inconsistent distribution of the features explains the high FAD score of 28.30 for synthetic audios of the "Bronchiectasis" class synthesized by MLP-VAE. Also, the low FAD score of 3.16 for the synthetic LRTI audios generated by MLP-VAE can be explained by the similarity between the features of the original and synthetic audios of the LRTI class. The distribution of features of audio segments generated by CNN-VAE shown in Fig 25 is consistent w.r.t. to the original audios compared to Conditional VAE. The features of these synthetic audios resemble those of the original audios better. Hence, the FAD score achieved by the synthetic samples of CNN-VAE is lower than that of MLP-VAE for certain classes. The spread of features of synthetic samples created by Conditional VAE shown in Fig 26 is inconsistent w.r.t the original audios for respiratory classes such as LRTI, URTI, Bronchiectasis and Bronchiolitis. Hence, synthetic audios created by CNN-VAE resemble the real audios more closely in a two-dimensional space when compared to MLP-VAE and Conditional VAE.

**7.1.3 Cross-correlation between synthetic and real audios.** Cross-correlation is a measure of how much two signals resemble each other [49]. The higher the correlation, the better is the similarity between the two signals. For 1D time-series $x$ and $y$, the cross-correlation at lag $k$, denoted by $\phi_{xy}[k]$, is given by the Eq (3).

$$\phi_{xy}[k] = \sum_{n=-\infty}^{\infty} x[k+n]y[n] \tag{3}$$

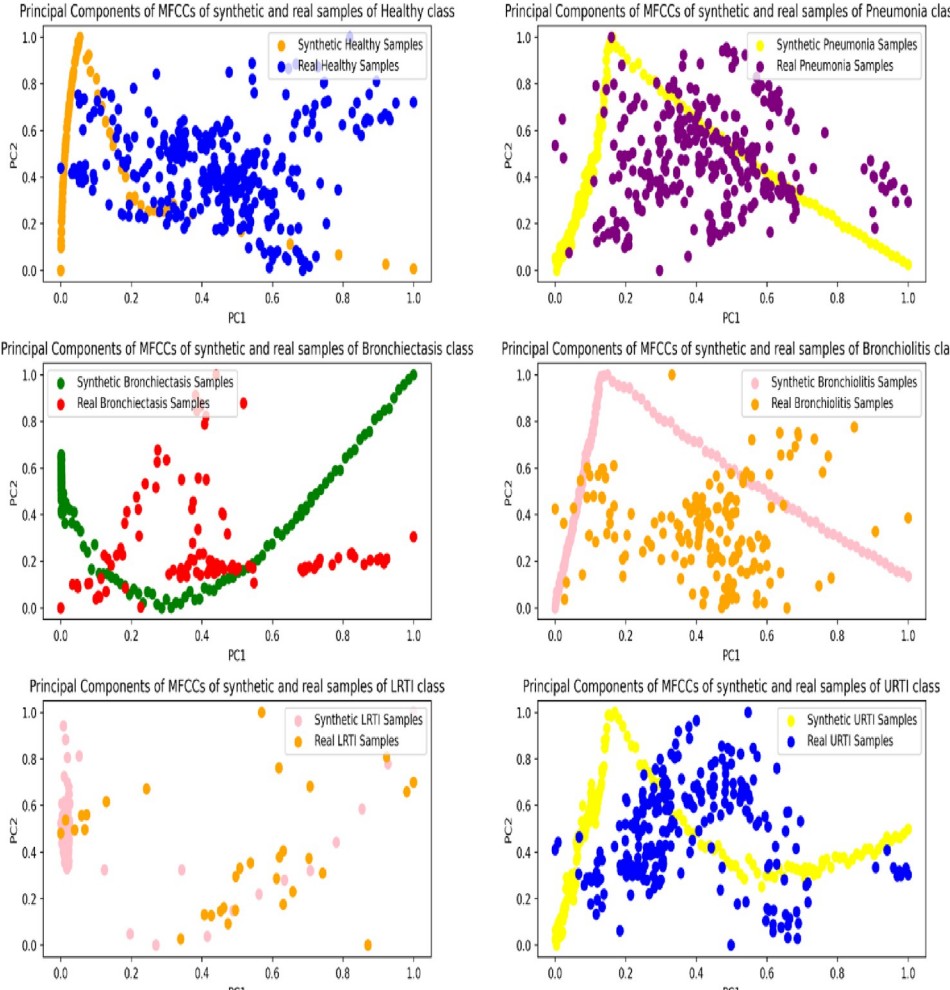

**Fig 24. Principal components of MFCCs of synthetic (MLP-VAE) and real samples of minority classes.**

From Eq (3), we observe that the cross-correlation operation multiplies a signal $y[n]$ with a shifted version of the signal $x[n]$ by introducing a lag $k$. This study used Scipy's correlate method to measure the cross-correlation between the real and synthetic audio segments. Since the real and synthetic signals were of the same length and the 'valid' mode argument of Scipy's correlate [50] method was used, we essentially measured the correlation between the real and synthetic signals at lag 0. To estimate the cross-correlation of the synthetic signals w.r.t. the real signals, we took a sample size of 50 from each minority class. The cross-correlation of each of the sampled synthetic signals w.r.t. the sampled real signals were measured, and the maximum correlation for each synthetic sample was selected. The mean and standard deviation of the maximum attainable correlation for each of the sampled synthetic samples w.r.t. the real samples are reported in Table 6. The correlation heatmaps in Figs 27–29 visualize the mean maximum attainable cross-correlation between the sampled synthetic samples of a given class and all minority classes.

The correlation heatmap in Fig 27 shows that the synthetic audio segments created by MLP-VAE are very weakly correlated with the real audio segments for all the classes except LRTI. The mean cross-correlation between the sampled synthetic LRTI signals created by MLP-VAE and the real LRTI signals is 0.79, the highest correlation compared to the

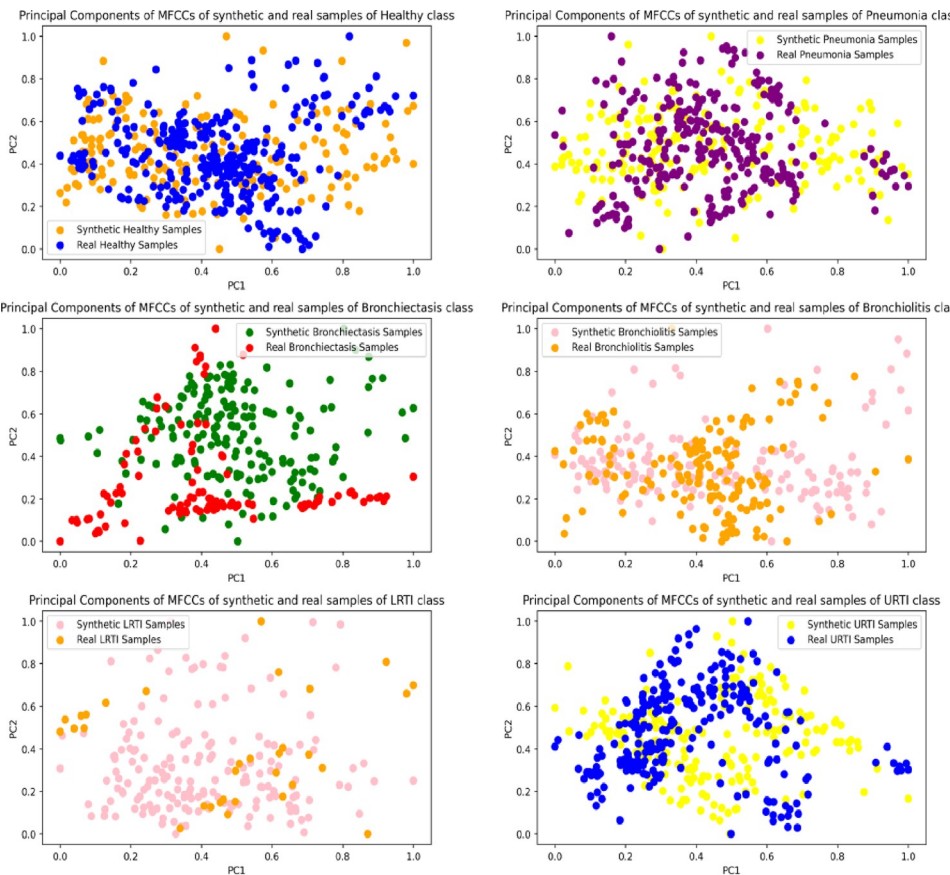

**Fig 25. Principal components of MFCCs of synthetic (CNN-VAE) and real samples of minority classes.**

corresponding correlations achieved by CNN-VAE (0.46) and Conditional VAE (0.50). This observation is consistent with our finding of MLP-VAE achieving the lowest FAD score for the LRTI class compared to CNN-VAE and Conditional VAE.

The correlation heatmaps in Figs 28 and 29 show that the synthetic audio signals created by CNN-VAE and Conditional VAE have a weak positive correlation w.r.t. synthetic samples. The figures show that the synthetic audio segments of a given class show the highest mean correlation w.r.t the real audio segments of a different class. In the case of Conditional VAE, the mean cross-correlations between the real audio segments of a given class w.r.t. the synthetic segments of all classes lie in a very narrow range. For example, the mean cross-correlation between the sampled real Bronchiectasis audio segments and that of synthetic segments lie in the range of 0.41–0.48, which implies that the synthetic samples belonging to the other classes that resemble Bronchiectasis can make it harder for our classifiers to classify the intended minority class correctly. Another example observed in Fig 28 is the high correlation of 0.65 between the synthetic Bronchiectasis samples and the real Pneumonia samples, which can cause the classifiers to misclassify real Bronchiectasis samples as Pneumonia, potentially hurting the performance of our classifiers with the Conditional VAE augmentation. A potential explanation for such observations could be that the features of the wheezes and crackles observed for a minority class such as Bronchiectasis are very similar to those observed in another minority class such as Pneumonia.

**7.1.4 Mel Cepstral Distortion.** The Mel Cepstral Distortion (MCD) measures the difference between two sequences of mel cepstra. To account for the differences in timing, we used

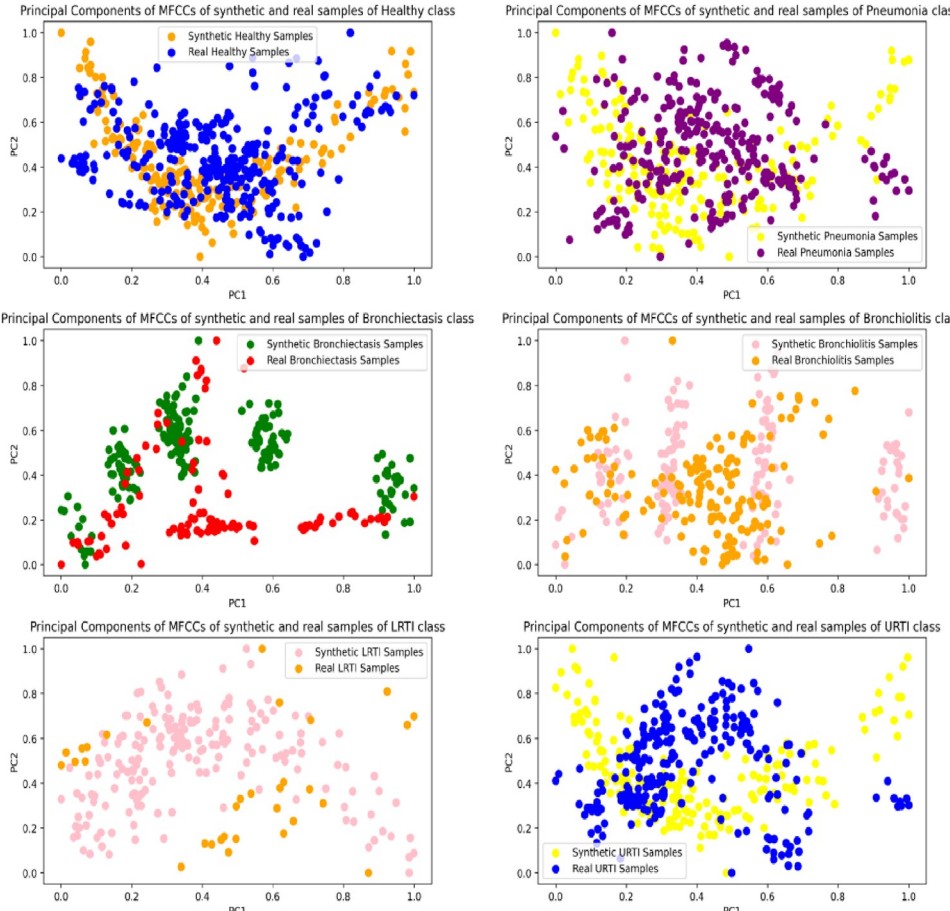

**Fig 26. Principal components of MFCCs of synthetic (Conditional VAE) and real samples of minority classes.**

Dynamic Time Warping (DTW) to identify the best possible alignment between the real and synthetic audio signal. We measured the difference MCD ($C_{ti}$, $\hat{C}_{ti}$) between the mel frequency cepstral coefficients (MFCCs) of the real $C_{ti}$ and synthetic $\hat{C}_{ti}$ audio segments over T time-frames using Eq (4)

$$\mathrm{MCD}\left(C_{ti},\ \hat{C}_{ti}\right) = \frac{10\sqrt{2}}{\ln 10}\frac{1}{T}\sum_{t=1}^{T}\sqrt{\sum_{i}\left(C_{ti} - \hat{C}_{ti}\right)^2} \qquad (4)$$

To measure the MCD between the real and synthetic signals of each class, we took samples of size 50 from each class and computed the MCD between each synthetic sample and all the

**Table 6. Cross-correlation between sampled synthetic and real audio segments for each class.**

| Class | MLP-VAE | CNN-VAE | Conditional VAE |
|---|---|---|---|
| Bronchiectasis | 0.1 ± 0.13 | 0.40 ± 0.15 | **0.46** ± 0.20 |
| Bronchiolitis | 0.34 ± 0.18 | 0.52 ± 0.12 | **0.57** ± 0.13 |
| Healthy | 0.01 ± 0.16 | **0.54** ± 0.15 | 0.52 ± 0.13 |
| LRTI | **0.79** ± 0.18 | 0.46 ± 0.15 | 0.50 ± 0.17 |
| Pneumonia | 0.32 ± 0.17 | 0.55 ± 0.14 | **0.56** ± 0.13 |
| URTI | 0.17 ± 0.15 | **0.59** ± 0.13 | 0.39 ± 0.15 |

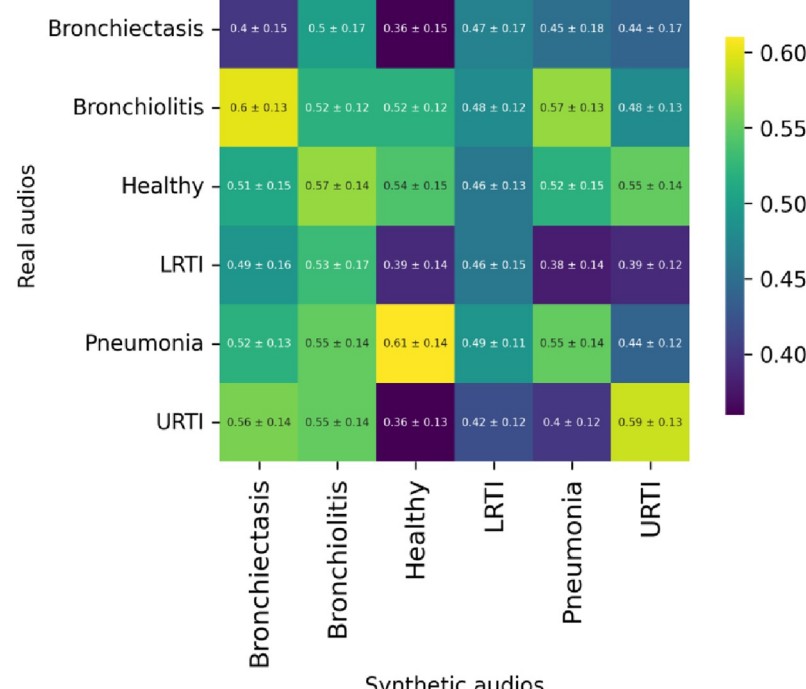

**Fig 27. Correlation heatmap between sampled synthetic (MLP-VAE) and real audio segments for all minority classes.**

**Fig 28. Correlation heatmap between sampled synthetic (CNN-VAE) and real audio segments for all minority classes.**

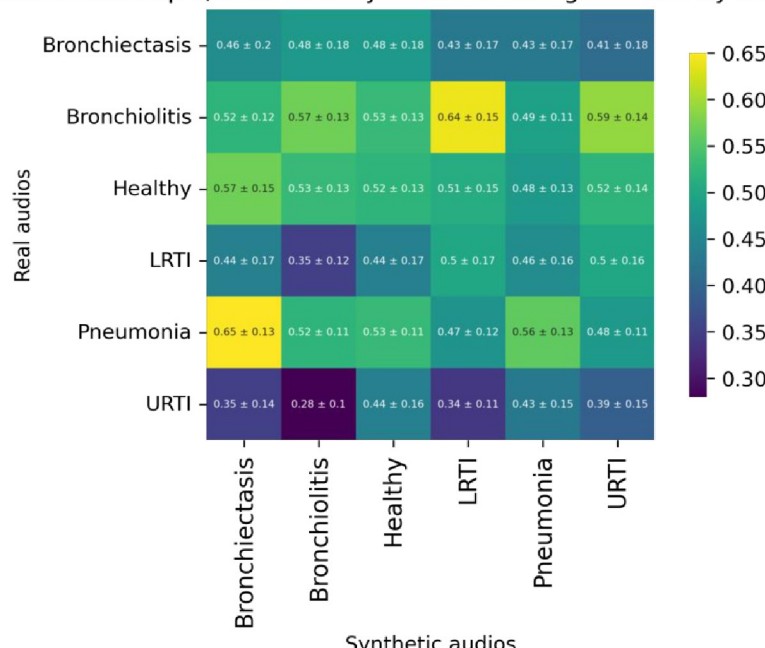

**Fig 29. Correlation heatmap between sampled synthetic (Conditional-VAE) and real audio segments for all minority classes.**

real audio segments of that class. The mean and standard deviation of these MCDs for a given class and VAE model are reported in Fig 30. Higher the MCD, the larger the distortion between the two mel cepstras and, hence, poorer the synthetic signal quality.

Fig 30 reveals that the MCD between the real and synthetic samples created by MLP-VAE and CNN-VAE for all classes except the Healthy class are consistent with each other. For the

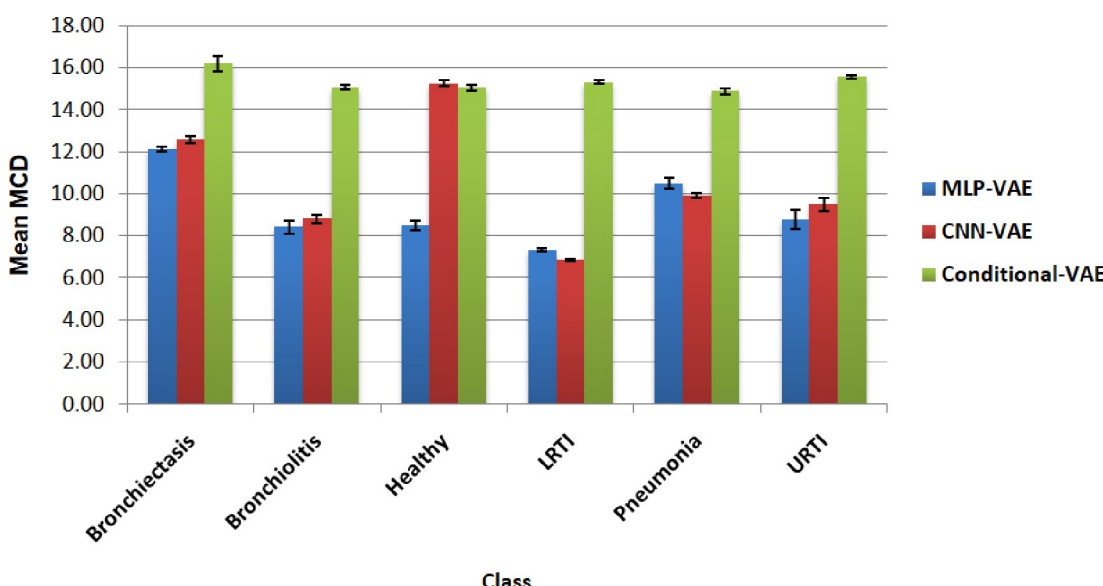

**Fig 30. Mean Mel Cepstral Distortion between the mel cepstras of the synthetic and real audio samples for all classes.**

synthetic samples created by Conditional VAE, the MCD is significantly higher, which can be justified because the Conditional VAE model was trained on all six minority classes simultaneously, giving it lesser time to learn the essential features of the disease classes.

## 7.2 Classification models

Various evaluation metrics used to determine the performance of a classification model are as follows:

1. **Confusion Matrix**: It provides an insightful idea of the classes that are predicted correctly and incorrectly by the model and a count of errors (False Positives and False Negatives) made. It is useful for measuring other metrics like recall, precision, specificity and accuracy. Fig 31 shows the structure of a confusion matrix and the commonly used terms with a confusion matrix.

2. **Precision**: Precision (Eq 5) represents the fraction of positive predictions that were actually correct. Hence, it is also called positive predictive value.

$$Precision = \frac{TP}{TP + FP} \tag{5}$$

3. **Recall**: Recall (Eq 6) represents the fraction of positive samples that were classified correctly. It is also known as sensitivity or true positive rate.

$$Recall = \frac{TP}{TP + FN} \tag{6}$$

4. **F1-Score**: F1 score (Eq 7) is the harmonic mean of precision and recall and is denoted by the formula

$$F1\ score = 2 \times \frac{Precision \times Recall}{Precision + Recall} \tag{7}$$

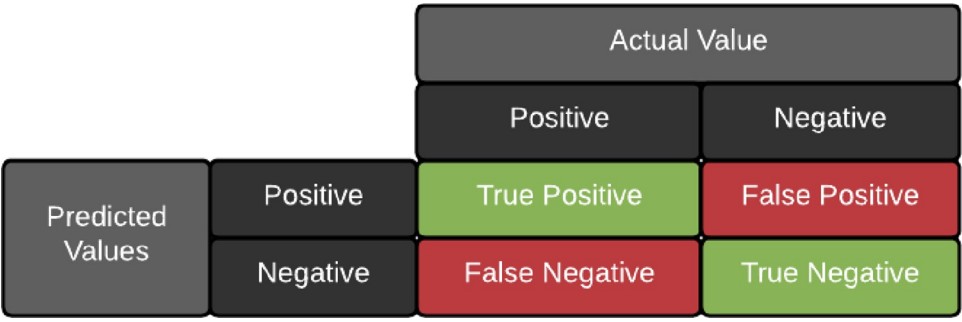

**Fig 31. Confusion matrix.**

5. **Specificity**: Specificity (Eq 8) represents the fraction of negative samples that were classified correctly. Hence, it is also called the true negative rate.

$$Specificity = \frac{TN}{TN + FP} \tag{8}$$

As indicated in Figs 33 to 37, the classification models can classify only COPD correctly and overfit for the minority classes with an imbalanced training set. This issue is resolved for certain minority classes with the help of our proposed data augmentation techniques. Table 7 shows the average of the classification metrics over all classes, whereas the confusion matrix gives us the actual performance of the classifiers per respiratory class. We trained each classifier on the four training sets (imbalanced + 3 augmented) thrice for 100 epochs. We reported the aggregated metrics in Table 6 and the best confusion matrix obtained from the three trials. To understand the impact of data augmentation on the classification models, consider the LRTI class, which is almost completely misclassified by classifiers trained on the imbalanced training set. In contrast, the classifiers classify LRTI better on the augmented training sets. With the augmented training sets, the misclassification proportion for the LRTI class was reduced from 100% for all classifiers (imbalanced) to 40% for ANN (CNN-VAE augmentation), 50% for CNN (CNN-VAE augmentation), 50% for LSTM (CNN-VAE augmentation), 50% for Resnet-50 (For all three VAEs) and 30% (Conditional VAE augmentation). The improvement in the performance metrics for the LRTI class after augmentation can also be observed in Fig 32, which compares the mean F1 score of the classifiers on the test set per class. Fig 32 for the LRTI class reveals that most of the classifiers (except LSTM and Resnet-50) completely misclassified LRTI with an imbalanced training set achieving a mean F1 score of 0; which improved to a minimum F1 score of 0.47 for MLP-VAE augmentation (except ANN), 0.37 for CNN-VAE augmentation (except ANN) and 0.52 for Conditional VAE augmentation (except ANN and CNN).

**Table 7. Impact of VAE augmentation on the performance of classification models.**

| Dataset | Metric | MLP | CNN | LSTM | RESNET-50 | EFFICIENT NET B0 |
|---|---|---|---|---|---|---|
| **Imbalanced training set** | **Specificity** | 0.95±0.09 | 0.96±0.06 | 0.92±0.16 | 0.97±0.05 | 0.94±0.11 |
| | **Sensitivity** | 0.48±0.3 | 0.29±0.37 | 0.41±0.28 | 0.75±0.2 | 0.47±0.32 |
| | **Precision** | 0.43±0.35 | 0.34±0.39 | 0.32±0.28 | 0.62±0.25 | 0.37±0.31 |
| | **F1 score** | 0.43±0.32 | 0.3±0.36 | 0.34±0.26 | 0.64±0.19 | 0.39±0.31 |
| **Augmented training set-1(MLP-VAE)** | **Specificity** | 0.97±0.05 | 0.96±0.08 | 0.92±0.15 | 0.98±0.05 | 0.96±0.07 |
| | **Sensitivity** | 0.51±0.29 | 0.61±0.2 | 0.41±0.24 | 0.71±0.16 | 0.55±0.22 |
| | **Precision** | 0.53±0.32 | 0.74±0.18 | 0.49±0.23 | 0.78±0.14 | 0.67±0.2 |
| | **F1 score** | **0.51±0.3** | **0.66±0.18** | **0.44±0.23** | **0.74±0.14** | **0.59±0.2** |
| **Augmented training set-2(CNN-VAE)** | **Specificity** | 0.95±0.11 | 0.96±0.07 | 0.92±0.17 | 0.98±0.04 | 0.96±0.06 |
| | **Sensitivity** | 0.45±0.33 | 0.62±0.23 | 0.38±0.26 | 0.71±0.14 | 0.56±0.26 |
| | **Precision** | 0.58±0.33 | 0.76±0.17 | 0.46±0.23 | 0.77±0.19 | 0.62±0.25 |
| | **F1 score** | **0.48±0.31** | **0.65±0.2** | **0.41±0.24** | **0.72±0.15** | **0.57±0.24** |
| **Augmented training set-3 (Conditional VAE)** | **Specificity** | 0.96±0.07 | 0.96±0.05 | 0.91±0.19 | 0.98±0.04 | 0.96±0.07 |
| | **Sensitivity** | 0.42±0.34 | 0.4±0.33 | 0.36±0.27 | 0.7±0.17 | 0.55±0.25 |
| | **Precision** | 0.48±0.33 | 0.48±0.35 | 0.52±0.23 | 0.76±0.17 | 0.6±0.23 |
| | **F1 score** | 0.41±0.32 | **0.39±0.3** | **0.42±0.24** | **0.72±0.15** | **0.56±0.23** |

Note: The improvement in F1 scores after augmentation are marked in bold.

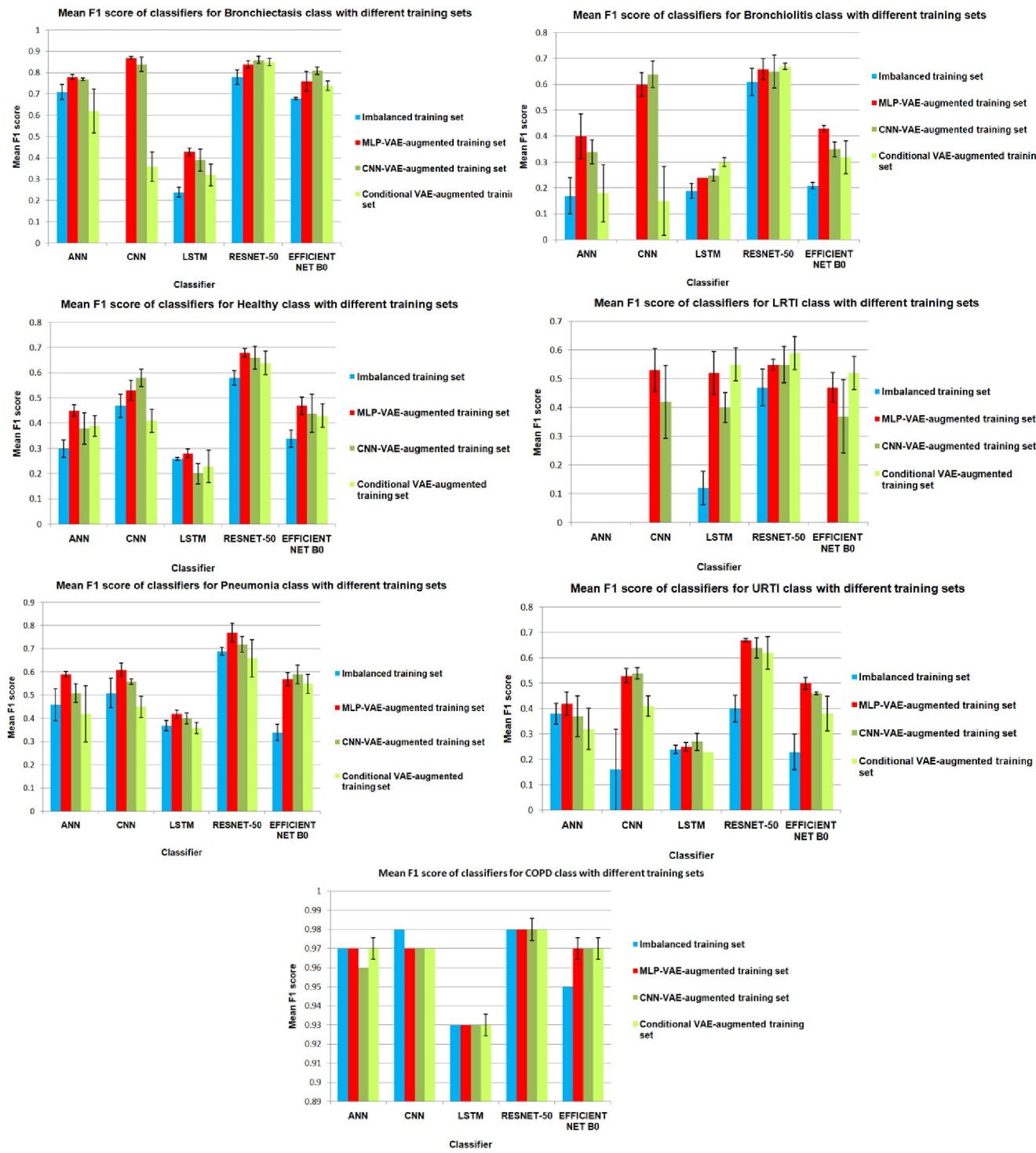

**Fig 32. Classwise comparison of F1 score achieved by the classifiers with different training set.**

On observing the performance of the classifiers on the augmented training sets in Fig 32, it can be seen that the MLP-VAE and CNN-VAE augmentations were more effective than Conditional VAE. The confusion matrices in Figs 33–37 for augmented training set 3 (Conditional VAE) show poor performance for all classifiers except Resnet-50 and Efficient Net B0

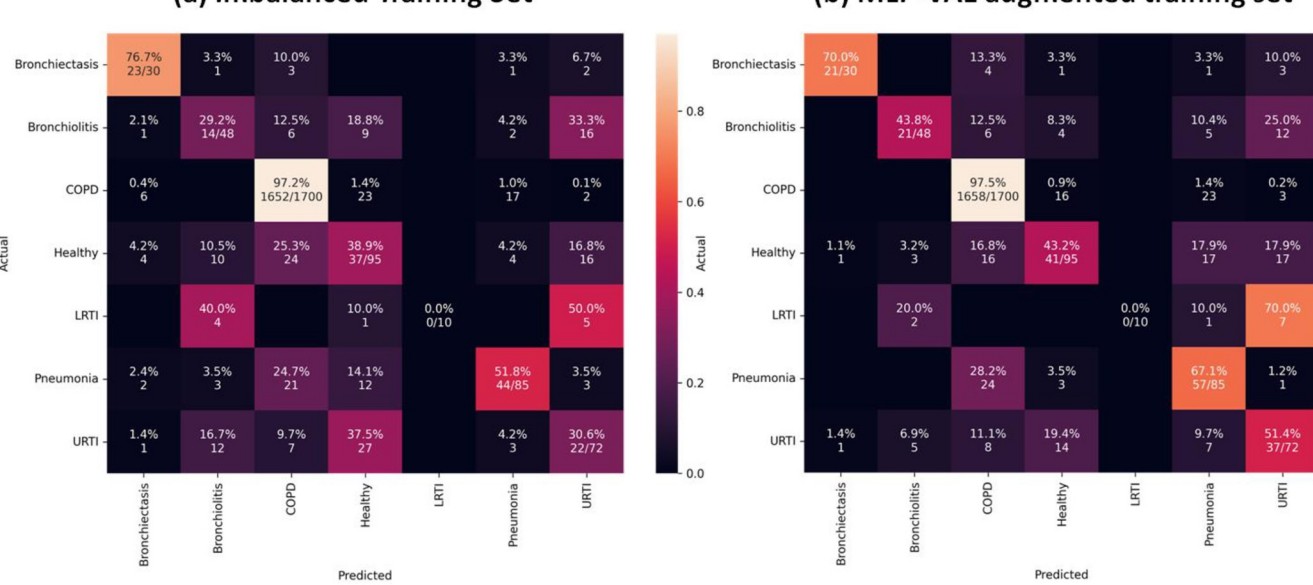

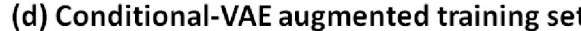

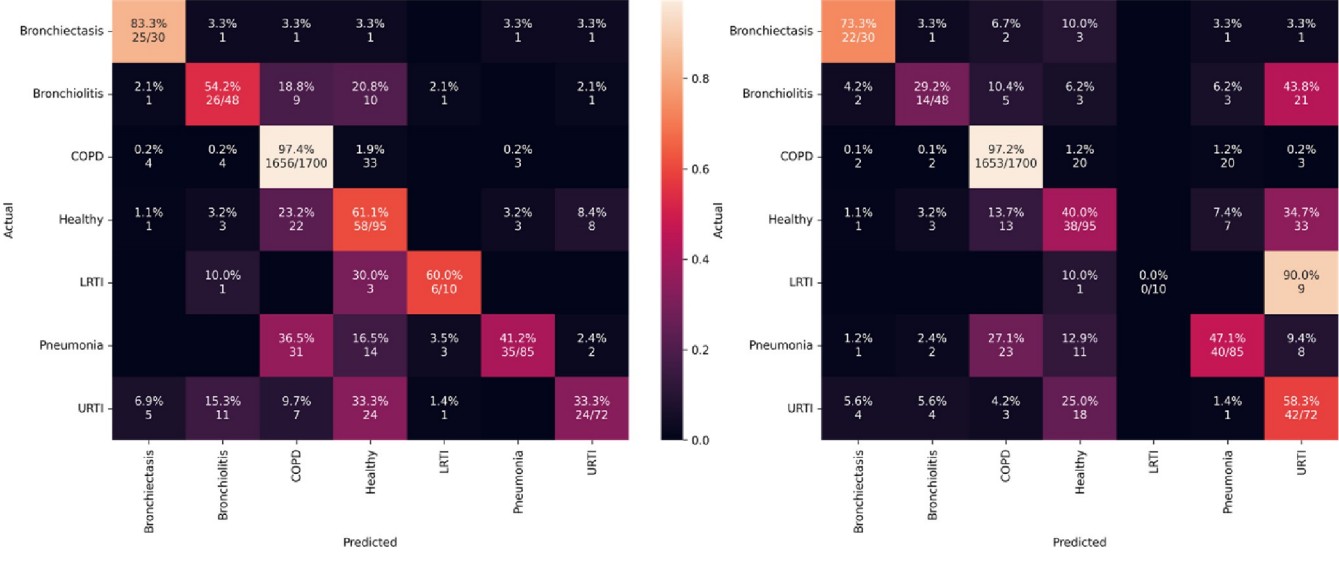

**Fig 33. Confusion matrices for ANN classifier with imbalanced and augmented training sets.**

compared to augmented training sets 1 (MLP-VAE) and 2 (CNN-VAE). The PCA plots in Fig 26 show that the distribution of features of the synthetic samples created by Conditional VAE disagrees with those of the original samples for LRTI, URTI, Bronchiectasis and Bronchiolitis classes. The synthetic samples generated by Conditional VAE create confusion for the classification models and hampers their performance compared to the augmentation of MLP-VAE and CNN-VAE.

On comparing the VAE augmentations, we can conclude that the MLP-VAE and CNN-VAE augmentations were more effective in enhancing the classification performance metrics than Conditional-VAE. Our evaluation of the generative models can justify this

**Table 8. Statistical significance of performance metrics achieved by various classifiers with imbalanced and augmented training sets.**

| Performance Metric | F ratio | Critical F value | p-value |
|---|---|---|---|
| Specificity | 0.13 | 2.63 | 0.94 |
| Precision | 9.01 | 2.63 | $8.72 \times 10^{-6}$ |
| Recall | 5.97 | 2.63 | 0.0005 |
| F1 score | 9.52 | 2.63 | $4.35 \times 10^{-6}$ |

observation, which revealed issues with the Conditional VAE model, such as the resemblance of the synthetic audios of a given class with another minority class and higher MCD of the synthetic audios than the MLP-VAE and CNN-VAE.

We used Excel's "two-way ANOVA (Analysis of Variance) with Replication" for comparing the performance metrics on the imbalanced and augmented training sets. The null hypothesis $H_0$ assumed for the test is "No significant difference in the performance metrics was observed with the augmented training sets", and the alternate hypothesis $H_1$ is "A significant difference in the performance metrics was observed with the augmented training sets". Table 8 reports the statistical test results for comparing performance metrics of various classifiers on the imbalanced and augmented training sets. The p-value represents the probability of the null hypothesis $H_0$ being true. A significance level (SL) of 0.05 was chosen for these tests. Table 8 shows that the p-values corresponding to the above hypothesis tests for all performance metrics were less than the chosen SL. Also, the F ratio was greater than the critical F value for all performance metrics except specificity. These test results indicate that the null hypothesis $H_0$ can be rejected, and a significant difference in the performance metrics is observed with the augmented training sets.

## 8. Discussion

This study aimed to evaluate the impact of augmenting the imbalanced ICBHI dataset with the synthetic samples created by VAEs to achieve superior classification results for detecting multiple respiratory diseases. Our results showed that the synthetic audio segments created by

**Table 9. Comparison of our results with recent works undertaken towards multi-class respiratory disease classification.**

| Authors and Year | Dataset used | Features / Input to model | Proposed Model(s) | Sensitivity | Specificity | ICBHI Score |
|---|---|---|---|---|---|---|
| [25] | ICBHI Dataset with healthy and **two classes** (Chronic and Non-Chronic) | MFCCs combined with their first-order derivative | LSTM | 0.98 | 0.82 | 0.90 |
| [30] | ICBHI Dataset with CNN VAE generated synthetic samples of **healthy** and **five disease classes** (Bronchiectasis, Bronchiolitis, COPD, Pneumonia, URTI) | Mel Spectrograms of respiratory sounds | CNN | 0.99 | 0.99 | 0.99 |
| [47] | ICBHI Dataset with augmented samples of **two classes** (COPD and Non-COPD) | MFCCs | CNN | 0.92 | 0.92 | 0.92 |
| [48] | King Abdullah University Hospital + ICBHI Database with **six classes** (Normal, COPD, BRON, Pneumonia, Asthma, heart failure) | Entropy-based features | Boosted Decision Trees | 0.95 | 0.99 | 0.97 |
| **Our Study (2021)** | ICBHI dataset with VAE-generated synthetic samples of **healthy and six disease classes** (Pneumonia, LRTI, URTI, Bronchiectasis, Bronchiolitis, COPD) | MFCCs of respiratory sound segments | MLP | 0.97 | 0.51 | 0.74 |
| | | | CNN | 0.96 | 0.62 | 0.79 |
| | | | LSTM | 0.92 | 0.41 | 0.67 |
| | | | RESNET-50 | 0.98 | 0.71 | 0.85 |
| | | | EFFICIENT NET B0 | 0.96 | 0.56 | 0.76 |

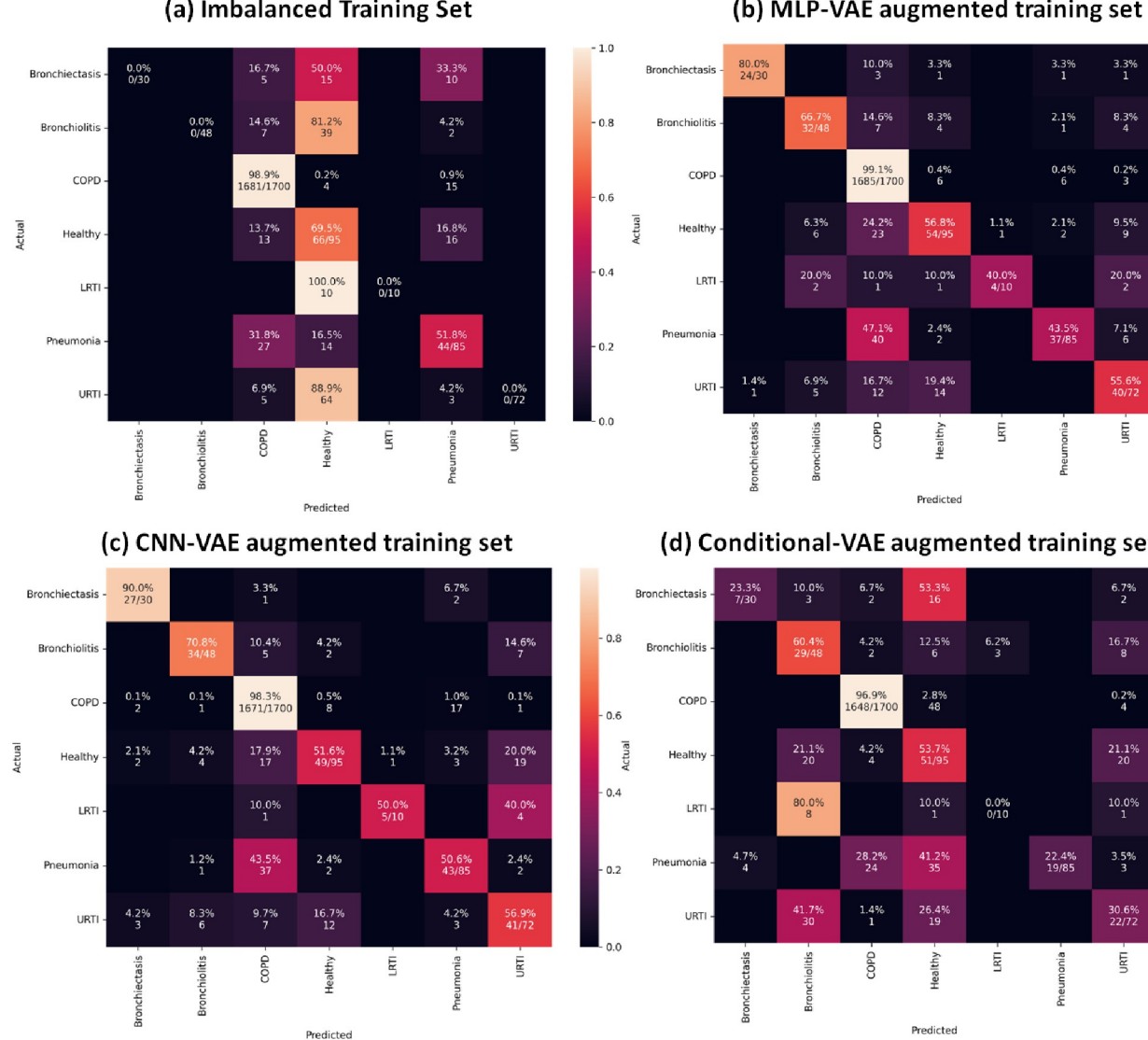

**Fig 34. Confusion matrices for CNN classifier with imbalanced and augmented training sets.**

MLP-VAE and CNN-VAE enhanced the classification performance metrics more than Conditional VAE. The VAE augmentation helped our classifiers achieve superior performance in detecting multiple diseases. In this section, we compared the results of our classifiers with those of existing respiratory disease classification models. We observed that very few works proposed models for the classification of multiple respiratory diseases in the literature. It is worth mentioning that a direct quantitative comparison of results is not possible due to the variations in datasets (sample size and disease classes involved), feature extraction techniques, classification algorithms and evaluation methods. Table 9 and Fig 38 collectively present the comparative summary of our work with recent works undertaken towards diseased respiratory sounds classification. Table 9 reveals that the sensitivity of our models is lower than those achieved by existing works. However, it is worth noting that the sensitivity metric in our case includes six disease classes, unlike other works where only two or four classes have been considered.

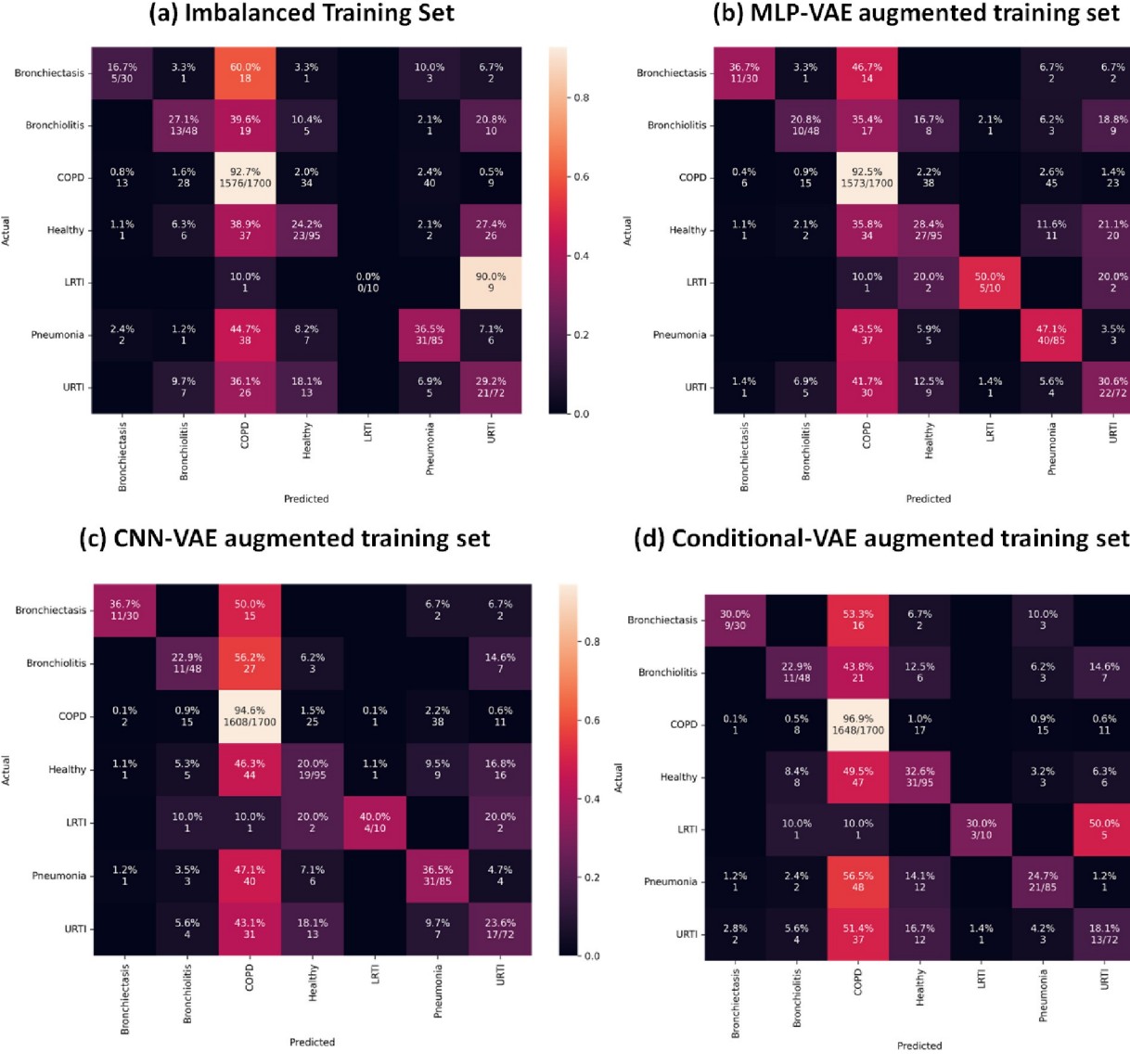

**Fig 35. Confusion matrices for LSTM classifier with imbalanced and augmented training sets.**

[25] proposed a deep learning-based auscultation framework consisting of RNN and its variants for detecting abnormal respiratory sounds (such as crackles, wheezes or both) and pathological lung diseases (such as chronic, non-chronic and healthy). Their results showed that MFCCs computed with a window step size leading to a 50% overlap between successive time frames lead to an LSTM with the best performance (ICBHI Score: 0.90, Sensitivity: 0.98 and Specificity: 0.82). Even though our work has not focused on classifying adventitious respiratory sounds such as wheezes and crackles, we have built models that can predict a specific disease rather than only its category (chronic or non-chronic). This was made possible due to the voluminous synthetic data we generated using VAEs to help our classifiers generalize better on the minority classes. Our proposed LSTM also extracted features from MFCCs of the respiratory sounds computed on overlapping time-frames; however, the extracted features were not sufficient for classifying the respiratory diseases with the imbalanced training set. Upon the VAE augmentation, the performance of our LSTM model improved significantly for certain disease

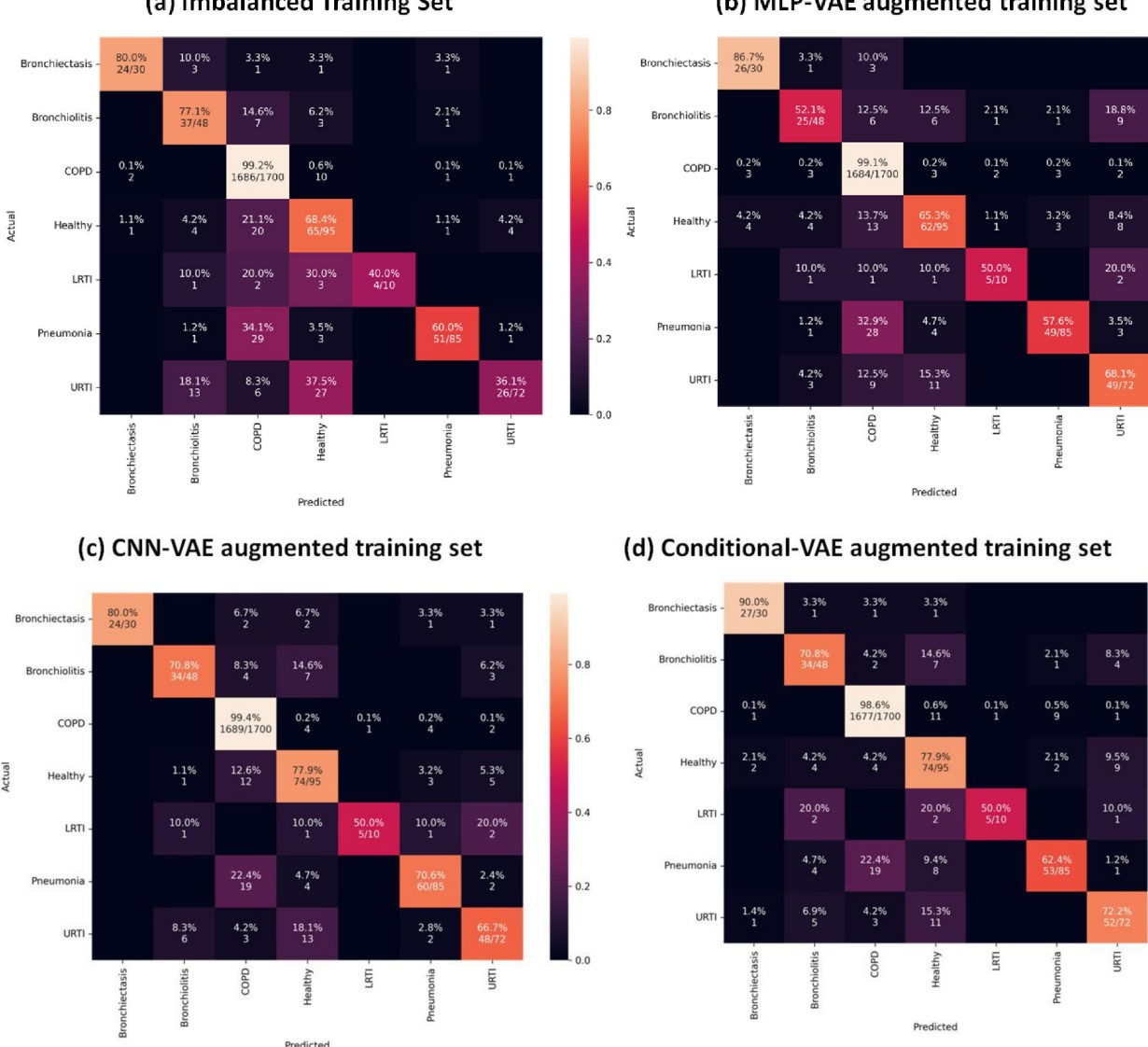

**Fig 36. Confusion matrices for RESNET-50 classifier with imbalanced and augmented training sets.**

classes and the Healthy class, achieving an ICBHI score of 0.67 (Sensitivity: 0.41 and Specificity: 0.92).

[30] developed 3-class and 6-class (healthy and five diseases) classification models for diagnosing respiratory diseases in the ICBHI dataset. Similar to our work, these authors proposed a CNN-VAE for augmenting the 6-class imbalanced dataset. The augmentation led to state-of-the-art results for their models with an ICBHI score of at least 0.99. Our work not only focused on 7-class (healthy + six diseases) classification models but also proposed a Conditional VAE, which allowed simultaneous training of the VAE for all minority classes. The disadvantage of CNN-VAE is that there is no way to control the output synthesized class, forcing us to train CNN-VAE for each respiratory class separately. Upon completing the training for a given class, the generative model can synthesize mel spectrograms for that class. As our classifiers rely on MFCCs, we needed to invert the synthetic mel spectrograms to raw audios. Training CNN-VAE for each class separately and inverting the synthetic mel spectrograms to raw

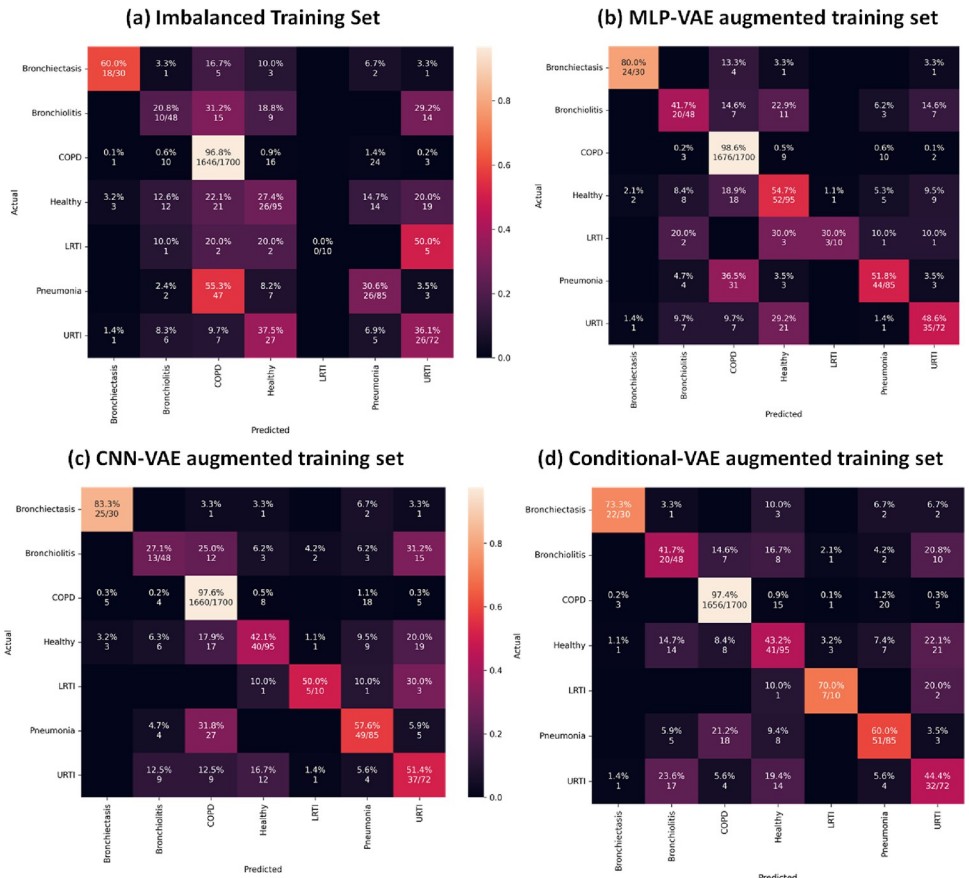

**Fig 37. Confusion matrices for Efficient Net B0 classifier with imbalanced and augmented training set.**

audios was time-consuming. To overcome this limitation of CNN-VAE, we experimented with a Conditional VAE, which could be guided via a label to generate mel spectrograms of specific classes, allowing us to train the VAE with all classes simultaneously. As a result, the time required for synthesizing the augmented training samples for all classes was reduced; however, our results showed that the Conditional VAE augmentation was harmful to the classifiers. Hence, we conclude that the unconditional models such as MLP-VAE and CNN-VAE, though time-consuming for training and generating synthetic samples, are suitable generative models for augmenting the respiratory sounds database.

[47] divided the ICBHI dataset into two classes consisting of COPD and non-COPD. They analyzed several audio representations such as MFCCs, Mel Spectrograms, Chroma STFT, Chroma CQT and Chroma CENS to identify the most effective audio representation for feature extraction using a CNN. They discovered that MFCCs and Mel Spectrograms led to the highest ICBHI scores of 0.77 and 0.73. They further improved these scores to 0.92 and 0.82 by augmenting their training set using augmentation techniques such as loudness augmentation, shift augmentation, mask augmentation and speed augmentation.

[48] proposed using entropy-based features such as Shannon entropy, logarithmic energy entropy and spectral entropy for classifying audio segments of multiple respiratory diseases. They used ensemble techniques such as bagging and boosting on classifiers such as Decision Tree and Linear Discriminant Analysis to enhance performance metrics such as sensitivity and specificity over all classes.

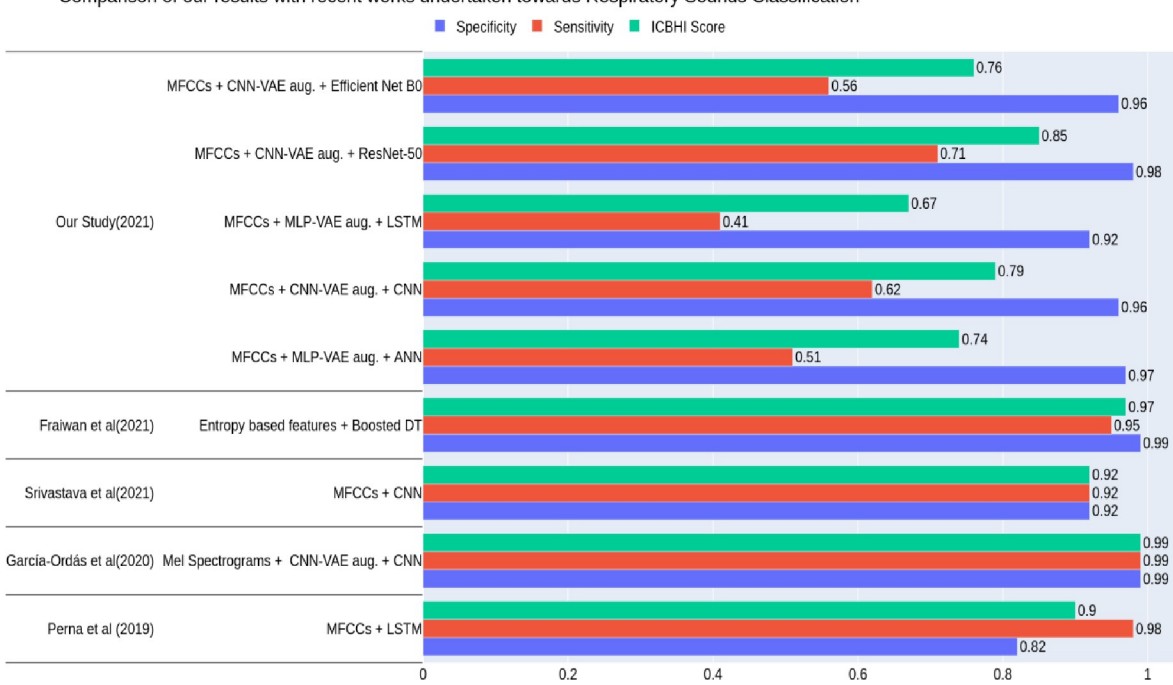

**Fig 38. Comparative summary of recent works undertaken towards respiratory sounds classification.**

## 9. Conclusion

To conclude, this paper investigated the impact of augmenting the imbalanced ICBHI dataset with synthetic audio segments created by VAEs on various classifiers. The resemblance between the features of the synthetic audio segments and that of the original was measured using metrics such as Fréchet Audio Distance, Cross-Correlation and Mel Cepstral Distortion. The results showed that the unconditional generative models such as MLP-VAE and CNN-VAE were more effective in enhancing the performance metrics of the classifiers than the Conditional VAE. The PCA plots for CNN-VAE indicated that generated audio segments also exhibited better variety in terms of acoustic features when compared to those created by MLP-VAE. Lastly, the classifiers completely misclassified certain diseases such as LRTI when trained on an imbalanced training set. Upon augmenting the imbalanced training set, a significant boost in the performance metrics of the classifiers was observed for certain minority classes and marginal improvement for the other classes.

## Author Contributions

**Conceptualization:** Shruti Patil, Satish Kumar.

**Data curation:** Jane Saldanha.

**Methodology:** Jane Saldanha, Shaunak Chakraborty.

**Project administration:** Shruti Patil.

**Supervision:** Shruti Patil, Ketan Kotecha, Anand Nayyar.

**Validation:** Shruti Patil, Satish Kumar, Anand Nayyar.

**Writing – original draft:** Jane Saldanha, Shaunak Chakraborty.

**Writing – review & editing:** Shruti Patil, Ketan Kotecha, Satish Kumar, Anand Nayyar.

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
