## [Decision Letter · Decision Letter 0]

1 Oct 2021

PONE-D-21-27133Variational Autoencoders for Audio Based Data Augmentation to Enhance Respiratory Disease ClassificationPLOS ONE

Dear Dr. Patil,

Thank you for submitting your manuscript to PLOS ONE. After careful consideration, we feel that it has merit but does not fully meet PLOS ONE’s publication criteria as it currently stands. Therefore, we invite you to submit a revised version of the manuscript that addresses the points raised during the review process.

 I strongly suggest the authors to carefully address all the major and minor concerns raised by the reviewers in the revised version.  Please submit your revised manuscript by Nov 15 2021 11:59PM. If you will need more time than this to complete your revisions, please reply to this message or contact the journal office at plosone@plos.org. Please include the following items when submitting your revised manuscript:A rebuttal letter that responds to each point raised by the academic editor and reviewer(s). You should upload this letter as a separate file labeled 'Response to Reviewers'.A marked-up copy of your manuscript that highlights changes made to the original version. You should upload this as a separate file labeled 'Revised Manuscript with Track Changes'.An unmarked version of your revised paper without tracked changes. You should upload this as a separate file labeled 'Manuscript'.

We look forward to receiving your revised manuscript.

Kind regards,

Murugappan M, Ph.D

Academic Editor

PLOS ONE

Journal Requirements:

2. Please amend your Methods section to include URLs (not just references) to where the data sets used in the study can be accessed.

“No,The funders had no role in study design, data collection and analysis, decision to publish, or preparation of the manuscript.”

4. Thank you for stating the following in the Funding Section of your manuscript:

“This work was supported by Symbiosis International (Deemed University).”

We note that you have provided funding information within the funding section that is not currently declared in your Funding Statement. Please note that funding information should not appear in the Acknowledgments section or other areas of your manuscript. We will only publish funding information present in the Funding Statement section of the online submission form.

“No,The funders had no role in study design, data collection and analysis, decision to publish, or preparation of the manuscript.”

Reviewers' comments:

Reviewer's Responses to Questions

**Comments to the Author**

1. Is the manuscript technically sound, and do the data support the conclusions?

Reviewer #1: Yes

Reviewer #2: Yes

2. Has the statistical analysis been performed appropriately and rigorously? 

Reviewer #1: Yes

Reviewer #2: Yes

3. Have the authors made all data underlying the findings in their manuscript fully available?

Reviewer #1: Yes

Reviewer #2: Yes

4. Is the manuscript presented in an intelligible fashion and written in standard English?

Reviewer #1: Yes

Reviewer #2: Yes

5. Review Comments to the Author

Reviewer #1: This paper discusses the application of deep neural VAE models to improve classification performance on diagnosing health conditions using recorded lung sounds from patients. The primary application of the VAE is the generation of additional samples for minority classes to overcome the overall skew of the dataset. Towards this goal, the authors investigate multiple augmentation strategies - MLP VAE, CNN VAE, and conditional VAE.

My main concerns with the paper are two fold -

1) Reproducibility and reuse - The key contribution of the paper lies in the precise augmentation methods described in section 6.3. I commend the authors for including architectural details, however, I am still unclear on how the conditional VAE was applied to the generation problem. What was the conditioning factor for generation? The second question I have is, how did the authors control the number of samples generated per condition from the unconditioned models in Table 3? When we run the unconditioned model, the samples may not be generated in the right ratio and may continue to be imbalanced.

So for point 1 - the authors need to discuss precisely how each model was run on the dataset, what was learned, what was the data generation strategy from the conditional and unconditional models, how these two models differed in terms of execution.

2) Originality - The concept of applying data augmentation strategies to most types of data (including audio) is fairly well explored. But more importantly, a few cited references such as [20] have considered similar directions as the proposed work. I would like to see a more thorough analysis of the knowledge gaps and shortcomings of the earlier papers that are bridged by the current study.

These two points should be satisfactorily addressed before the paper is in a publishable form.

Reviewer #2: Some concerns that need to be addressed:

1. Though the work address the problem of imbalanced signal data for respiratory disease diagnosis, the performance analysis is not clear. Data augmentation can only be used for training data. The validation should be done only on original test data.

2. The similarity of the augmented signal and original signal should be analyzed even more rigorously for classes for which the available data is very much limited namely for classes.

3. The data augmentation leads to only marginal increase in signal classification accuracy. Justify?

6. PLOS authors have the option to publish the peer review history of their article (what does this mean?). If published, this will include your full peer review and any attached files.

Reviewer #1: **Yes: **Adit Krishnan

Reviewer #2: **Yes: **Palani Thanaraj

---

## [Author Response · Author response to Decision Letter 0]

3 Jan 2022

Original Manuscript ID: PONE-D-21-27133

Original Article Title: Variational Autoencoders for Audio Based Data Augmentation to Enhance Respiratory Disease Classification

To: PLOS ONE Editor

Re: Response to reviewers

Dear Editor,

Thank you for allowing a resubmission of our manuscript, with an opportunity to address the reviewers’ comments.

We are uploading (a) our point-by-point response to the comments (below) (response to reviewers), (b) an updated manuscript with yellow highlighting indicating changes, and (c)a clean updated manuscript without highlights (PDF main document).

Best regards,

Jane Saldanah,Shaunak Chakraborty,Shruti Patil ,Satish Kumar, Ketan Kotecha and Anand Nayyar

Reviewer#1, Concern # 1: The authors need to discuss precisely how each model was run on the dataset, what was learned, what was the data generation strategy from the conditional and unconditional models, how these two models differed in terms of execution.

Author response: Thank you for your kind suggestion.

Author action: We have added explanation about the training procedure and data generation strategy for the VAE models in section 6.3 (highlighted in yellow).

Reviewer#1, Concern # 2: The concept of applying data augmentation strategies to most types of data (including audio) is fairly well explored. But more importantly, a few cited references such as [20] have considered similar directions as the proposed work. I would like to see a more thorough analysis of the knowledge gaps and shortcomings of the earlier papers that are bridged by the current study.

Author response: Thank you for your suggestions.

Author action: We have added two paragraphs on page number 10 in section 4 / Research Gaps describing the novelties of our work and the issues we noticed in previous papers, such as [30], that have been addressed in this study. 

Reviewer#2, Concern # 1: Though the work address the problem of imbalanced signal data for respiratory disease diagnosis, the performance analysis is not clear. Data augmentation can only be used for training data. The validation should be done only on original test data.

Author response: Thank you for the suggestion.

Author action: We have added the details of the train and test sets used for building the VAEs and classification models in section 3 / Exploratory Data Analysis (highlighted in yellow). The system architecture diagram has also been updated to give a clearer picture of the performance evaluation of the classifiers. As per the suggestion received, the synthetic data was used for augmenting the training set only. Hence, we re-evaluated the performance of the classifiers on the hold-out test set, containing real audio samples only. The updated performance metrics, including the confusion matrices, statistical analysis and other graphs have been reported in the manuscript.

Reviewer#2, Concern # 2: The similarity of the augmented signal and original signal should be analyzed even more rigorously for classes for which the available data is very much limited, namely for classes.

Author response: Thank you for your kind suggestion.

Author action: We have further elaborated our contributions in sections 7.1.3 and 7.1.4. We have measured the cross-correlation and Mel cepstral distortion between the real and synthetic samples for all minority classes to elaborate the similarity of the synthetic samples w.r.t the real samples.

Reviewer#2, Concern # 2: The data augmentation leads to only a marginal increase in signal classification accuracy. Justify?

Author response: Thank you for the suggestion.

Author action: On observing the performance metrics of the classifiers aggregated over all classes, it does seem like the augmentation had a little improvement. However, upon comparing the performance metrics for the minority classes with the imbalanced and augmented training sets, a significant difference can be observed. For example, minority classes such as LRTI were being completely misclassified with the imbalanced training set ; post augmentation a significant improvement in the performance metrics can be observed.

---

## [Decision Letter · Decision Letter 1]

22 Mar 2022

Data Augmentation Using Variational Autoencoders for Improvement of Respiratory Disease Classification

PONE-D-21-27133R1

Dear Dr. Patil,

We’re pleased to inform you that your manuscript has been judged scientifically suitable for publication and will be formally accepted for publication once it meets all outstanding technical requirements.

Kind regards,

Murugappan M, Ph.D

Academic Editor

PLOS ONE

Additional Editor Comments (optional):

Reviewers' comments:

Reviewer's Responses to Questions

**Comments to the Author**

1. If the authors have adequately addressed your comments raised in a previous round of review and you feel that this manuscript is now acceptable for publication, you may indicate that here to bypass the “Comments to the Author” section, enter your conflict of interest statement in the “Confidential to Editor” section, and submit your "Accept" recommendation.

Reviewer #1: All comments have been addressed

2. Is the manuscript technically sound, and do the data support the conclusions?

Reviewer #1: Partly

3. Has the statistical analysis been performed appropriately and rigorously? 

Reviewer #1: Yes

4. Have the authors made all data underlying the findings in their manuscript fully available?

Reviewer #1: Yes

5. Is the manuscript presented in an intelligible fashion and written in standard English?

Reviewer #1: Yes

6. Review Comments to the Author

Reviewer #1: 1) The authors have presented a fairly thorough analysis of the three variational autoencoder models leveraged for data augmentation for the minority classes - MLP, CNN, and Conditional VAEs. The generated synthetic samples are then leveraged to train a varied selection of classification models and measure the efficacy of the data augmentation strategy on the overall performance metrics.

2) The authors provide a limited analysis of the quality of the generated synthetic samples. What is the impact of the VAE architecture on these parameters? Although the authors have provided architectural details, I suspect that precise overall reproducibility of the performance metrics may not be possible owing to the stochasticity of the VAE models. Although a nearly matching performance should be achievable with similar steps as described by the authors.

3) On the whole, this paper provides a fairly effective data augmentation strategy to deal with skew in audio datasets. The strategy is intuitive and can be applied to other classes, datasets, and domains in audio analysis. Thus, I recommend an Accept for this paper. However, I highly encourage the authors to include additional analyses of the model architectures and provide detailed explanations for the observed results. For instance, a technical analysis of the problems associated with the conditional VAE (which appears to underperform the other two VAE architectures in this case) could be helpful to apply this research to other datasets.

7. PLOS authors have the option to publish the peer review history of their article (what does this mean?). If published, this will include your full peer review and any attached files.

Reviewer #1: **Yes: **Adit Krishnan

---

## [Editor Report · Acceptance letter]

4 Aug 2022

PONE-D-21-27133R1 

Data Augmentation Using Variational Autoencoders for Improvement of Respiratory Disease Classification 

Dear Dr. Patil:

I'm pleased to inform you that your manuscript has been deemed suitable for publication in PLOS ONE. Congratulations! Your manuscript is now with our production department. 

Kind regards, 

on behalf of

Dr. Murugappan M 

Academic Editor

PLOS ONE